# Dissolved organic matter thiol concentrations determine methylmercury bioavailability across the terrestrial-marine aquatic continuum

Emily Seelen [1,2] ✉, Van Liem-Nguyen [3], Urban Wünsch [4], Zofia Baumann [1], Robert Mason [1], Ulf Skyllberg [5] & Erik Björn [3] ✉

The most critical step for methylmercury (MeHg) bioaccumulation in aquatic food webs is phytoplankton uptake of dissolved MeHg. Dissolved organic matter (DOM) has been known to influence MeHg uptake, but the mechanisms have remained unclear. Here we show that the concentration of DOM-associated thiol functional groups (DOM-RSH) varies substantially across contrasting aquatic systems and dictates MeHg speciation and bioavailability to phytoplankton. Across our 20 study sites, DOM-RSH concentrations decrease 40-fold from terrestrial to marine environments whereas dissolved organic carbon (DOC), the typical proxy for MeHg binding sites in DOM, only has a 5-fold decrease. MeHg accumulation into phytoplankton is shown to be directly linked to the concentration of specific MeHg binding sites (DOM-RSH), rather than DOC. Therefore, MeHg bioavailability increases systematically across the terrestrial-marine aquatic continuum as the DOM-RSH concentration decreases. Our results strongly suggest that measuring DOM-RSH concentrations will improve empirical models in phytoplankton uptake studies and will form a refined basis for modeling MeHg incorporation in aquatic food webs under various environmental conditions.

It is estimated that anthropogenic mercury (Hg) emissions have caused a three-fold increase in Hg content in surface marine waters compared to pre-industrial conditions[1], increasing the threat of Hg to ecosystem viability and human health[2]. Methylmercury (MeHg) is of particular concern for human health as it is a neurotoxin that bioaccumulates and biomagnifies in the aquatic food web[3], potentially causing high human exposure via fish consumption[4,5]. The largest enrichment step of MeHg into the aquatic food web occurs in the processes of cellular uptake of MeHg from water by seston. For this step, bioconcentration factors (BCF) range from 3 to 7 log units[6], which

is several orders of magnitude greater than the approximate ten-fold increase in MeHg concentration with each successive trophic level[3,7]. Uptake by unicellular organisms such as small algae and bacteria has been shown to control MeHg concentrations in fish[6] and thereby the potential exposure of MeHg to humans.

Dissolved organic matter (DOM) is ubiquitous in aquatic systems and plays several critical roles in the biogeochemical cycling of Hg[8]. Here, DOM refers to all dissolved organic compounds in a natural system, while dissolved organic carbon (DOC) refers to a quantitative measure of the C content of DOM. At environmentally relevant

[1]University of Connecticut, Department of Marine Sciences, Groton, CT, USA. [2]University of Southern California, Earth Sciences, Los Angeles, CA, USA. [3]Umeå University, Department of Chemistry, Umeå, Sweden. [4]Technical University of Denmark, National Institute of Aquatic Resources, Section for Oceans and Arctic, 2800 Lyngby, Denmark. [5]Swedish University of Agricultural Sciences, Department of Forest Ecology and Management Umeå, Umeå, Sweden. ✉ e-mail: seelen@usc.edu; erik.bjorn@umu.se

concentrations, MeHg associated with DOM or particulate OM is bound to thiol (RSH) groups present in the matter[9,10], but challenges in RSH measurements have led to DOC being the most common proxy used to represent the concentration of dissolved organic MeHg-binding ligands. In laboratory experiments, the cellular uptake of MeHg by phytoplankton generally decreases with increased concentrations of DOC[11,12], and several field studies have shown a negative correlation between DOC and MeHg BCF[6,13]. Such relationships indicate that DOC controls MeHg BCFs in natural systems[6] by reducing MeHg's bioavailability to phytoplankton[14]. However, the effect of DOC on MeHg uptake tends to vary with DOM source and properties[11,15–20] and exceptions have been reported where naturally low DOC waters led to lower accumulation of MeHg in plankton than naturally high DOC waters[21]. Particularly large variability in MeHg bioconcentration has been reported for different DOM types at DOC concentrations less than ~5 mg/L[15]. Primarily, these differences have been attributed to unique DOM characteristics including binding site abundances, structure, and aromaticity[11,12,16].

Studies more closely focused on the uptake of specific MeHg complexes have shown how the chemical speciation of dissolved MeHg can influence its cellular uptake[12,22,23] in phytoplankton cultures. Through such studies, both passive uptake of MeHgCl and active uptake of MeHg complexes with low molecular mass thiols[12,23] have been supported, and the uptake rate is generally faster for inorganic MeHg complexes, and particularly MeHgCl, relative to MeHg bound to organic ligands including DOM[24]. Thermodynamic stability and sterical hindrance effects have been identified as factors controlling the

uptake rate and accumulation efficiency of specific MeHg complexes by phytoplankton[12,23]. Translating uptake potentials from relatively simple laboratory-based studies to complex natural conditions, however, is challenging and requires a mechanistic understanding of how DOM controls MeHg speciation between available ligands in unique aquatic systems. The differences in uptake rates and bioaccumulation mechanisms for MeHg bound to DOM versus inorganic ligands heightens the importance of adequately calculating MeHg speciation when predicting MeHg uptake into phytoplankton. An improved understanding of these processes is necessary for the prediction of biotic MeHg exposure now and under future environmental change scenarios[20,25].

In this study, we addressed critical aspects of MeHg bioavailability at two different scales using DOM isolated from a diverse range of coastal waters. First, we evaluated MeHg bioavailability in the presence of DOM based on a generalizable understanding of MeHg–DOM interactions. Second, we elucidated how and why MeHg bioavailability changes across four distinct regions of the terrestrial-marine aquatic continuum based on measured DOM properties. The goals of this study were based on the hypotheses that the DOM (i) binding capacity and (ii) binding affinity decrease, while (iii) the rate of ligand exchange for MeHg–DOM complexes increase, across the terrestrial-marine continuum. The bases for these hypotheses are outlined in the results and discussion. If true, such changes in DOM properties would contribute to an increased availability for cellular uptake and accumulation of MeHg by phytoplankton in marine compared to upstream waters. We evaluated how natural DOM controls MeHg bioavailability by a comprehensive characterization of thermodynamic (binding capacity and binding affinity) and kinetic (rate of ligand exchange reactions) properties of MeHg–DOM interactions. The DOM principles we highlight as driving MeHg uptake were then tested in a phytoplankton uptake experiment. Finally, we elucidated site-specific MeHg bioavailability using updated speciation models. Our results all point to the concentration of thiol ligands associated with DOM as the main control on MeHg bioavailability in natural waters.

## Results and discussion
### Characteristics of the study sites and DOM properties
Water column samples were collected from 12 unique estuaries and two offshore locations, 20 sites in total, along the northeast coast of the United States (38.9–44.7°N, Fig. 1, Fig. S1). The sites were chosen along estuarine salinity gradients (0.5–36 practical salinity units (psu), median = 21 psu) with DOC concentrations ranging from 116–590 μM (median = 235 μM) (Table 1). One highly contaminated site was included (Berry's Creek, NJ) leading to a large range in dissolved MeHg (0.001–2.78 pM, median = 0.067 pM) and total Hg (0.61–45 pM, median = 1.7 pM) concentrations in the sampled water (Table S1). DOM was extracted from 20–40 L of surface water at each site following Dittmar et al.[26]. This solid-phase extraction method (based on a styrene divinyl benzene polymer, so-called PPL-SPE) is the most common approach for DOM enrichment and desalting, which is a prerequisite for most types of molecular characterizations[27], including the ones in this study. In general, PPL-SPE captures a broad range of compounds and retains overall differences in the molecular composition of the DOM between samples[28–31]. However, since external factors such as sample salinity impact the carbon yield of the extraction, we normalized relevant parameters to DOC concentrations to avoid quantitative biases. The DOM samples were characterized by bulk C, N, and S composition, fluorescence spectral analyses[17,32], sulfur speciation (determined by S K-edge X-ray absorption near edge (XANES) spectroscopy), and total thiol content (RSH/DOC) (determined by liquid chromatography-tandem mass spectrometry, LC-MS/MS). For comparison, the sites were separated into four classes: marsh, river, estuary, and shelf (Table 1). Similarities and differences in DOM properties among the site classes were evaluated by principal component analysis (PCA).

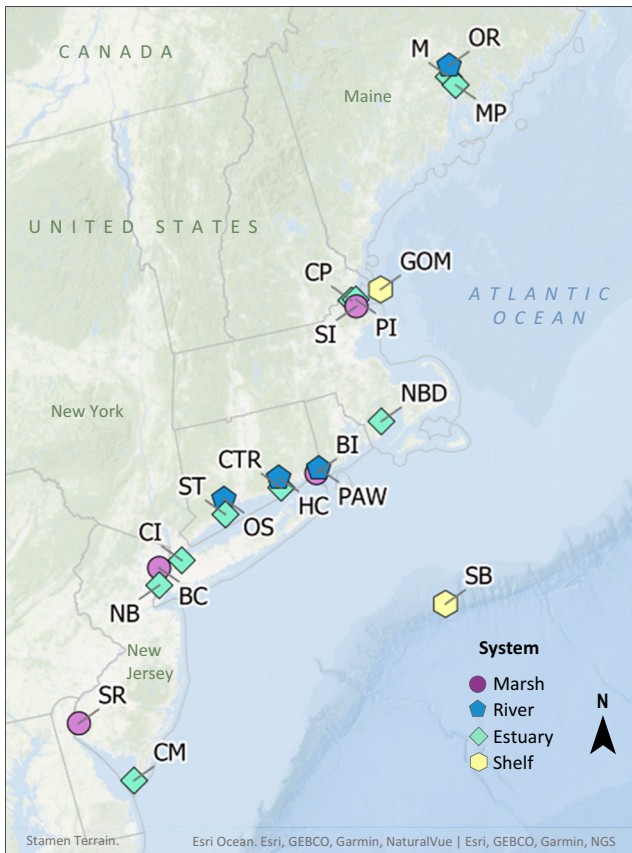

**Fig. 1 | Site map.** Location of the study sites along the northeastern coast of the United States. Maps were created in QGIS using the Ersi Ocean Basemap credited to Esri, GEBCO, Garmin, NaturalVue | Esri, GEBCO, Garmin, NGS, and the Stamen Terrain Background map tiles by Stamen Design, under CC BY 4.0. Data by OpenStreetMap, under ODbL. More details on the sites by site code are given in Table 1 and S1 and Fig. S1.

**Table 1 | Site and extracted DOM characteristics**

| Site characteristics | | | | | | | | Extracted DOM characteristics | | | | | | |
|---|---|---|---|---|---|---|---|---|---|---|---|---|---|---|
| Site | US state | System type | Latitude (degrees) | Longitude (degrees) | Salinity (psu) | DOC (µM) | DOM-RSH in water (nM) | % C | % N | % S | C/S (mol ratio) | C/N (mol ratio) | RSH/DOC (µmol/mol) | Log K[1] |
| SR | DE | Marsh | 39.58 | -75.48 | 4.0 | 289 | 71.0 | 45.6 | 1.5 | 1.6 | 76 | 37 | 246 +/- 11 | 16.2 |
| BC | NJ | Marsh | 40.83 | -74.08 | 4.7 | 590 | 215 | 52.5 | 2.1 | 2.0 | 69 | 29 | 364 +/- 23 | 16.8 |
| BI | CT | Marsh | 41.34 | -71.88 | 28 | 258 | 108 | 53.4 | 2.5 | 4.4 | 32 | 25 | 418 +/- 31 | 16.5 |
| SI | MA | Marsh | 42.75 | -70.84 | 33 | 553 | 187 | 52.6 | 1.9 | 2.0 | 70 | 32 | 338 +/- 18 | 16.3 |
| OS | CT | River | 41.31 | -73.09 | 0.6 | 245 | 26.1 | 49.1 | 1.9 | 0.9 | 140 | 30 | 107 +/- 6 | 16.3 |
| PAW | RI | River | 41.37 | -71.83 | 0.5 | 291 | 26.2 | 60.7 | 1.9 | 1.0 | 160 | 37 | 90 +/- 35 | 16.5 |
| HC | CT | River | 41.38 | -72.35 | 2.2 | 282 | 36.2 | 49.1 | 1.5 | 0.9 | 148 | 38 | 128 +/- 4 | 16.1 |
| OR | ME | River | 44.69 | -68.82 | 4.7 | 422 | 18.7 | 51.0 | 0.8 | 0.5 | 253 | 77 | 44 +/- 3 | 16.4 |
| CM | DE | Estuary | 38.94 | -74.97 | 32 | 134 | 15.5 | 50.1 | 2.2 | 2.0 | 65 | 27 | 115 +/- 11 | 16.4 |
| NB | NJ | Estuary | 40.67 | -74.13 | 22 | 236 | 23.0 | 47.2 | 2.1 | 2.4 | 52 | 26 | 97 +/- 9 | 16.7 |
| CI | NY | Estuary | 40.85 | -73.78 | 28 | 177 | 13.1 | 54.9 | 2.7 | 2.0 | 74 | 23 | 74 +/- 5 | 16.7 |
| ST | CT | Estuary | 41.17 | -73.11 | 18 | 206 | 34.8 | 39.4 | 1.5 | 1.2 | 88 | 30 | 169 +/- 7 | 17.0 |
| CTR | CT | Estuary | 41.29 | -72.35 | 21 | 191 | 13.6 | 49.9 | 2.0 | 1.1 | 125 | 30 | 71 +/- 4 | 16.2 |
| NBD | MA | Estuary | 41.65 | -70.91 | 32 | 194 | 14.3 | N/A | N/A | 1.4 | N/A | N/A | 74 +/- 28 | N/A |
| PI | MA | Estuary | 42.82 | -70.82 | 30 | 150 | 28.5 | 48.8 | 2.0 | 2.6 | 50 | 29 | 190 +/- 20 | N/A |
| CP | MA | Estuary | 42.82 | -70.88 | 9.7 | 233 | 11.4 | 55.6 | 2.1 | 1.3 | 113 | 31 | 49 +/- 3 | 17.4 |
| MP | ME | Estuary | 44.43 | -68.94 | 32 | 158 | 14.6 | 54.2 | 1.5 | 1.3 | 108 | 43 | 92 +/- 10 | N/A |
| M | ME | Estuary | 44.59 | -68.86 | 15 | 285 | 11.2 | 56.9 | 1.2 | 0.6 | 261 | 55 | 39 +/- 1 | 16.2 |
| SB | CT | Shelf | 39.84 | -70.74 | 36 | 116 | 5.1 | 53.1 | 2.2 | 0.7 | 212 | 28 | 44 +/- 5 | 16.6 |
| GOM | ME | Shelf | 42.85 | -70.48 | 32 | 117 | 8.9 | 50.2 | 1.9 | 1.4 | 96 | 31 | 76 +/- 8 | 16.6 |

[1]The specified Log $K$ values (uncertainty ±0.19) corresponds to the reaction Eq. (2).
Classification and location of the study sites and selected key parameters of the sites and extracted dissolved organic matter (DOM) samples. The site locations are within the United States and are shown in Fig. 1. System types were determined based on individual site features and local salinity, as defined in the text. The carbon (C), nitrogen (N), and sulfur (S) percentages were calculated by weight. Site specific thiol concentrations (DOM-RSH in water) were calculated by multiplying the thiol (RSH) to dissolved organic carbon (DOC) ratio (µmol/mol) by the in situ DOC concentration.

One PCA focused upon the fluorescence spectral data (Fig. 2a, b). The first principal component of this PCA described 61% of the data variability and separated samples containing more recently produced fluorescent material (C360) from those with relatively older material (C450). The second principal component described 27% of the data variability and discriminated fluorescent DOM by origin (C390 bacterially derived material, C520 terrestrially dominated material). The shelf site DOM samples stood out with higher proportions of recently produced material of lower molecular weight (inferred from the spectral slope ratio) than the other site classes. Class separation for the non-shelf sites was less obvious and suggests the main DOM source in these regions is terrestrial input. A slight watershed effect was observed in the PCA as sites from the same geographical system tended to group together (compare site locations in Fig. 1 with site grouping in Fig. 2a).

Another PCA was made to visualize the chemical speciation of dissolved organic sulfur by location given its role in MeHg binding (Fig. 2c, d). The first two principal components described 79% of the data variability (with 52% in the first principal component) and the sites were more strongly separated in this PCA compared to the one based on fluorescence spectral data. The greatest contributions to site separation came from the redox state of dissolved organic sulfur where the most oxidized forms, sulfate and sulfonate, and the most reduced forms, organic disulfides (RSSR) and RSH + organic mono-sulfides (RSR), loaded onto opposite sides of the plot. The XANES results demonstrated a characteristically high proportion of the three reduced organic sulfur forms (i.e., the sum of RSH, RSR, and RSSR) at the river and marsh sites, a high proportion of organic sulfate and sulfonate at shelf sites, and a large variability among the estuarine sites likely explained as a mixing gradient between the terrestrial and marine DOM sources (Fig. 2c, d). Overall, the fluorescence analysis and dissolved organic sulfur speciation results of the extracted DOM samples supported the initial geographical grouping of sites into the

four classes: marsh, river, estuary, shelf, and show that the sites represent a range in key parameters covering the terrestrial-marine aquatic continuum.

## Thermodynamics of the MeHg interaction with DOM − binding capacity and binding affinity

The binding capacity and binding affinity of MeHg ligands are important thermodynamic parameters influencing the chemical speciation and bioavailability of dissolved MeHg in natural waters. Here, we define the MeHg-binding capacity as the concentration of strong binding ligands, i.e., the thiol functional groups. The MeHg-binding capacity of the extracted DOM is expressed as the thiol content per mol carbon of the DOM (RSH/DOC, µmol/mol; here forward referred to as the DOM binding capacity), and the in situ MeHg-binding capacity in water at a given site is expressed as the total molar concentration of DOM-associated thiols (DOM-RSH, nmol/L; here forward referred to as the in situ binding capacity). The in situ binding capacity was determined by multiplying the DOM binding capacity by the local DOC concentration (mol/L). We quantified the DOM binding affinity as the formation constant (log $K$) of MeHg(DOM-RS) complexes for the different extracted DOM. The two thermodynamic parameters (RSH/DOC and log $K$) were determined using independent experiments. The DOM binding capacity was quantified using a recently developed thiol-specific method based on LC-MS/MS[33]. Binding affinity was determined with a competing ligand exchange reaction (adapted from Liem-Nguyen et al.[34]) using N-Acetyl-Penicillamine (Nacpen) as the competing ligand and specifically quantifying the MeHg(Nacpen) complex by LC-inductively coupled plasma MS (LC-ICPMS).

The DOM binding capacity varied by a factor of ~10 among the sites (range 39−418 µmol RSH/mol DOC) (Table 1) with the marsh sites having the highest RSH/DOC and marine sites the lowest, although the marine sites were not statistically different from the river and estuarine sites. The trend in RSH/DOC agreed with the organic reduced sulfur

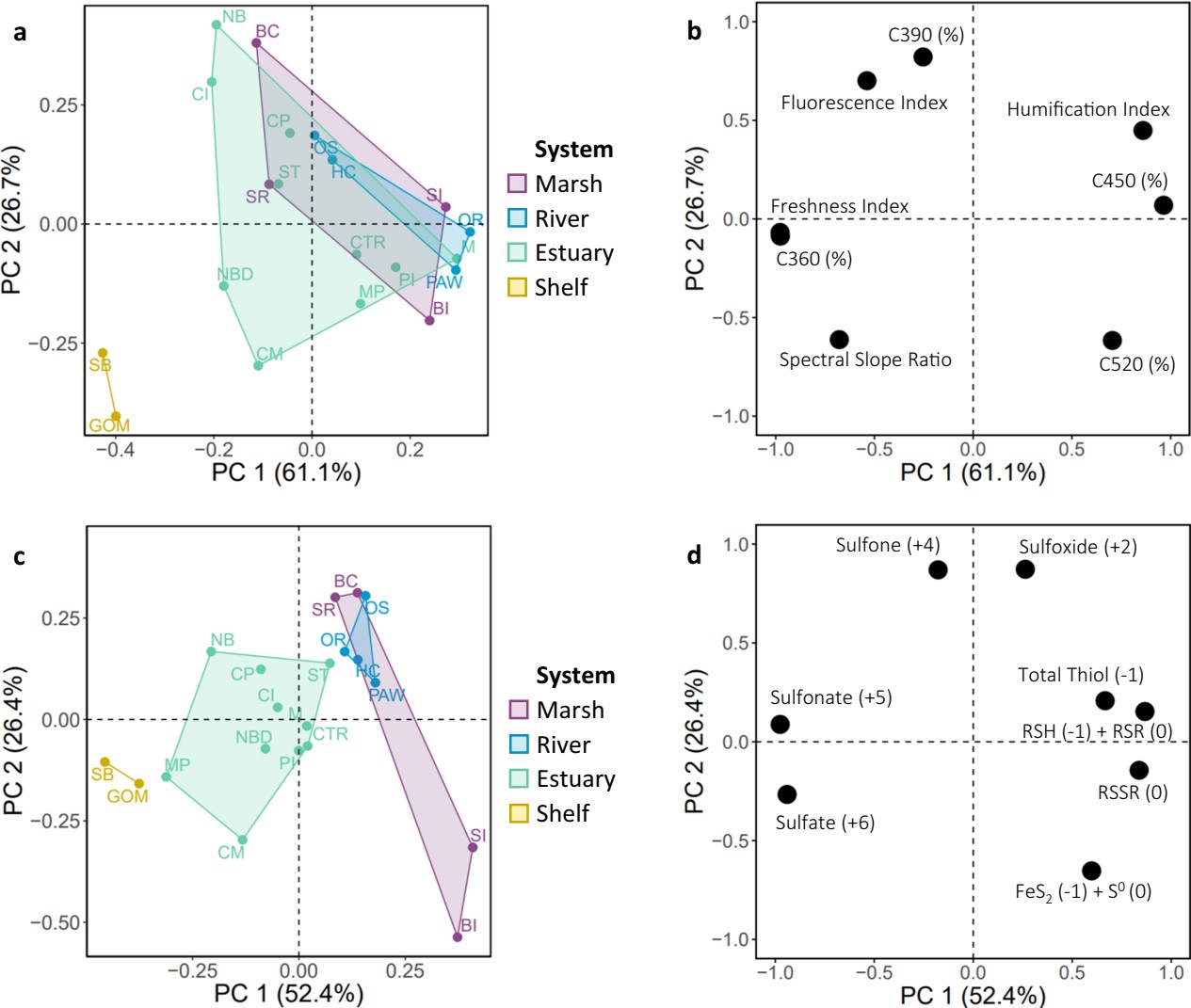

**Fig. 2 | Properties of dissolved organic matter among system types.** Principal component analyses score (**a**, **c**) and loading (**b**, **d**) plots illustrating the relationship between **a**, **b** fluorescence spectroscopy parameters and **c**, **d** S speciation results for each site (site codes are given in Table 1). Classification of sites is shown in the left score plot panels where the colored shapes represent unique systems and the sites are labeled as listed in Table 1. Variable loading plots are shown on the right. The formal oxidation state of the sulfur reference compounds, and the independently quantified total thiols, are shown in parenthesis in **d**. Source data are provided as a Source Data file.

pool of the dissolved organic sulfur as measured by S XANES, (RSH + RSSR)/DOC, and the two independent measurements corroborate a general decrease in reduced dissolved organic sulfur compounds, including thiols, from marshes and rivers into marine systems. The variability in the RSH/DOC ratio showed that DOC concentration alone is a poor proxy for the in situ MeHg-binding capacity across different aquatic ecosystems. Indeed, the in situ DOC concentrations varied by only a factor of five among sites (116–590 μM), but the in situ DOM-RSH concentration varied by a factor of 40 (5–215 nM). The DOM-RSH concentration varied systematically with the type of site where the average value decreased in the order: marsh > river ≈ estuary > shelf (Fig. 3a). The substantial range and systematic trend in DOM-RSH concentration across our study sites denote in situ binding capacity as a key factor controlling MeHg speciation and bioavailability across terrestrial-marine aquatic environments.

The thiol content of natural DOM has not been reported in many field or experimental lab studies, in part because their isolation and quantification are analytically challenging. To compare with the few published studies, we divided our independently determined RSH (by LC-MS/MS) by the organic reduced sulfur (RSSR + RSR + RSH by S XANES). The results ranged from 1.3% to 5.5% (average 2.8%), with the lowest fractions found at the marine shelf sites and the highest at marsh-impacted sites. This is lower than Suwannee River DOM (2R101N, from the International Humic Substances Society), which was determined to have ~15% RSH of organic reduced sulfur in two prior independent studies[33,35], as well as solid peat and wetland soils which have 15–30% RSH of organic reduced sulfur[9,34–36]. Indirect, operationally defined procedures that quantify undefined "strong binding sites" for MeHg and $Hg^{2+}$ in DOM (e.g., [37,38]) are less certain than the direct measures used in this study since they are sensitive to the experimental conditions[39] and specific ligand types are not defined. Even so, the general patterns of $Hg^{2+}$-binding sites per g DOC observed by Lamborg et al.[40] are similar to the trends in DOM binding capacity found here, with ratios decreasing from terrestrial to marine environments. The consistent trends across sites means that the RSH/DOC results in Table 1 provide a means to estimate in situ MeHg-binding capacities in coastal waters for which local DOC concentrations and site characteristics are known. Furthermore, the recently developed LC-MS/MS method used here to determine total thiol concentrations[33] is more accessible, and requires less sample amounts, than extended

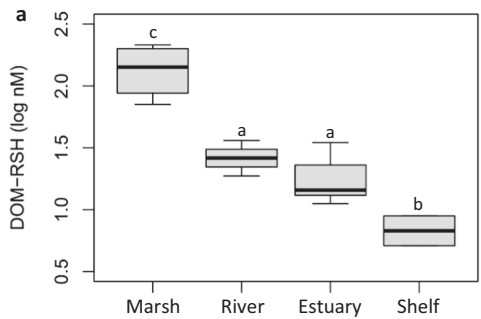
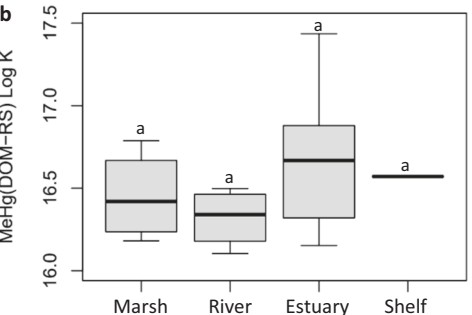

**Fig. 3 | System-specific DOM-RSH concentrations and MeHg(DOM-RS) stability constants. a** Total in situ dissolved organic matter thiol concentrations (log nM DOM-RSH) and **b** stability constant Log $K$ values for MeHg(DOM-RS) as described by Eq. (2) for each site grouping across the terrestrial to marine aquatic continuum. Significant differences among the site groupings are noted by the letters above the boxes. The number of sites in each group and the individual site data can be found in Table 1. The top and bottom edges of each box correspond to the first and third data quartiles, respectively, while the center line represents the data mean. Whiskers extend to the furthest data points. Source data are provided as a Source Data file.

X-ray absorption fine structure and XANES techniques and will facilitate an expansion of such measurements in future studies.

Regarding MeHg-binding affinities (log $K$) of the DOM samples, we hypothesized that they would reflect the strong bond between MeHg and thiol ligands (e.g.[9,37]). Even though MeHg dominantly binds with reduced thiol functional groups, the strength of the MeHg-S bond can be influenced by structural electrostatic effects, as demonstrated by Reid and Rabenstein[41]. For instance, protonation of amine functional groups located close to the thiol moiety can reduce the log $K$ of MeHg-thiol complexes by redistributing charge from the thiol moiety[41]. Because the C/N ratio is generally lower for marine algal DOM compared to terrestrial humic DOM[42], we hypothesized that the log $K$ would slightly, but systematically, decrease across the terrestrial to marine continuum.

We determined MeHg–DOM stability constants by the competing ligand exchange experiments based on specific quantification of the MeHg(Nacpen) complex by LC-ICPMS, the total DOM-RSH concentration, and the previously determined stability constant for the MeHg(Nacpen) complex[41]. The log $K$ for MeHg complexes with the unique DOM extracts varied from $16.1 \pm 0.19$ to $17.4 \pm 0.19$ (Eq. 2, Table 1, Fig. 3b). This range agrees well with previous studies on MeHg binding to reduced sulfur groups (presumed to be thiols) in aquatic[39] and soil[9,10,43,44] organic matter (Table S2). The average log $K$ of 16.5 is also close to that of the MeHg complex with Nacpen (log $K = 16.76$), the competing ligand in these experiments, and to other specific thiol compounds[41,45]. This result supports the hypothesis that MeHg binds solely to thiol groups in the DOM. A key finding, however, is that there was no systematic trend among the site classes even though log $K$ varied among the DOM extracts (Fig. 3b), and there was no correlation between log $K$ and the C/N ratio in contrast to the hypothesis. We thus conclude that MeHg-binding affinity is likely not a primary factor that systematically controls MeHg bioavailability across the terrestrial to marine aquatic continuum.

## Kinetics of MeHg ligand exchange with DOM

In addition to thermodynamics, the chemical speciation and bioavailability of MeHg can be kinetically controlled by sterical hindrance effects induced by the 3D structure of MeHg complexes[12,23]. In this way, large or "bulky" chemical structures may lead to a slower exchange rate of MeHg between dissolved and particulate phases, including cells and other ligands. It has been shown that substantial fractions of inorganic Hg(II) bound to DOM and mineral phases can form pools that exchange ligands at very low rates (so-called "non-exchangeable" pools)[46]. If MeHg formed such pools with natural DOM, bulk MeHg measurements would overestimate the bioavailable pool and underestimate MeHg BCFs. Most studies suggest fast ligand exchange kinetics for dissolved MeHg complexes[45,47,48], including with DOM[11],

but have lacked the ability to decipher non-exchangeable pools. We used a competing ligand exchange experiment where Nacpen and DOM-RSH were pre-equilibrated with separate enriched stable isotopes of MeHg, i.e., Me$^{204}$Hg(Nacpen) and Me$^{200}$Hg(DOM-RS). The two equilibrated isotopes were mixed, and the concentration and Hg isotopic composition of MeHg(Nacpen) determined starting at five minutes after mixing until steady-state was reached. Ultimately, the non-exchangeable pool of MeHg was small (<10%, Fig. S2) for all the investigated DOM samples and not significantly different under different DOM and competing ligand ratios. We thus conclude that formation of non-exchangeable MeHg pools is not a quantitatively important process for the interaction of MeHg with the highly contrasting DOM types investigated in this study.

Even though practically all MeHg was exchangeable, bioavailability and cellular uptake can still be influenced by the rate of ligand exchange. For the investigated shelf (LIS), estuary (NB) and river (PAW) DOM, MeHg exchanged between binding ligands faster than could be measured with the competing ligand exchange method, i.e., steady-state was reached before 5 min (Fig. S2). The two marsh DOM (SI and BI) exchange rates were somewhat slower, but still barely measurable. A slower ligand exchange rate for the marsh DOM is consistent with its larger molecular size distribution compared to the other site classes (Fig. S3). The larger molecular structure of marsh DOM, assuming DOM-RSH compounds are also larger, could in theory contribute to a lower cellular uptake rate of MeHg bound to marsh-type DOM-RSH compounds. However, the exchange rate of MeHg between the competing ligand Nacpen and all the investigated DOM samples (i.e., between MeHg(Nacpen) and MeHg(DOM-RS)) was considerably faster than the exchange rate between MeHg(Nacpen) and phytoplankton cells determined in a previous study with green algae[23]. Applying the same pseudo-first-order rate model as Skrobonja et al.[23], the turnover for MeHg exchange between Nacpen and DOM-RSH was 2–3 min for marsh sites and <1 min for the other DOM types in our study, which is much faster than the ~30 min turnover for MeHg exchange between Nacpen and green algae cells[23] at similar concentrations (100–300 nM) of Nacpen. These results show that the MeHg ligand exchange reactions among DOM-RSH compounds in solution are not a major process controlling differences in MeHg bioavailability across the terrestrial to marine aquatic continuum.

Hypothetically, the results of the kinetic exchange tests could be influenced by the use of extracted DOM if the molecular composition of a water sample impacts the extraction yields of specific compounds. While the PPL-SPE used in this study captures a broad range of compounds and retains overall differences in the molecular composition of the DOM, recent studies suggest that small hydrophilic molecules are partly lost from the sorbent during the extraction process[31]. Such compounds form the most bioavailable and easily exchanged MeHg-

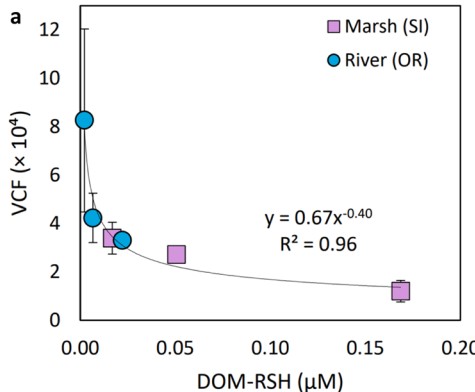
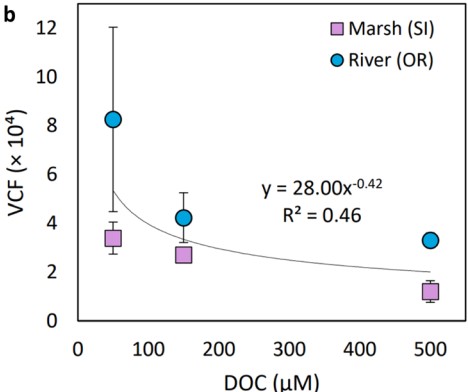

**Fig. 4 | Methylmercury accumulated in phytoplankton.** Phytoplankton MeHg volume concentration factors (VCF) for a marine diatom, *Thalassiosira pseudonana*, for two contrasting DOM types, one marsh site (SI) and one river site (OR). **a** depicts the VCFs versus the concentration of thiol ligands associated with dissolved organic matter (DOM-RSH). **b** shows the same VCF information but plotted against the concentration of dissolved organic carbon (DOC) used in the experiments. Error bars represent standard deviation ($n = 3$, except for site SI with lowest DOM-RSH and DOC concentration where $n = 2$) and are smaller than visible in some cases. Source data are provided as a Source Data file, including replicate VCF data.

thiol complexes[12,23]. In theory, if the recovery of such compounds were incomplete in the DOM extraction, the in situ turnover for MeHg ligand exchange might be even faster than the <1–3 min we calculated. Therefore, our conclusion that MeHg ligand exchange kinetics among DOM-RSH compounds is fast and not a major control of MeHg bioavailability would still be valid, or even amplified, if there were losses of small hydrophilic thiols in the PPL-SPE isolation.

## Bioavailability and VCF of MeHg depend on the total concentration of DOM-associated thiols

This study aimed to provide a molecular understanding for how DOM decreases MeHg bioavailability to phytoplankton. We showed through a series of thermodynamic and kinetic reaction experiments that the in situ MeHg-binding capacity of DOM varied significantly and systematically across the terrestrial-marine aquatic continuum, whereas binding affinity varied but not significantly or systematically, and ligand exchange kinetics were fast for all types of DOM. To test whether the absolute DOM-RSH concentration in water dictates MeHg bioaccumulation, and whether DOM-RSH concentration is a stronger predictor for MeHg BCFs than DOC concentration (as commonly used in previous studies), an uptake experiment was conducted. The diatom *Thalassiosira pseudonana* was exposed to MeHg pre-equilibrated with varied concentrations of DOM from one marsh (SI) and one river (OR) site with highly contrasting RSH/DOC ratios and overall sulfur speciation (Table 1 and Fig. 2b). These sites were chosen for their similar MeHg(DOM-RS) stability constants (log $K$ = 16.3 and 16.4, respectively) to specifically test the hypothesized causal relationship between DOM-RSH concentration and MeHg uptake. The results are presented as volume concentration factors (VCFs, which are BCFs normalized to the volume of the cells) after 4 hours of uptake (Fig. 4).

The measured VCFs are of similar magnitude to prior studies[11,49] and depict similar trends with DOC to those measured across a created DOC gradient by Luengen et al.[11] (Fig. S4). As hypothesized, the results showed a considerably stronger relationship between MeHg VCF and DOM-RSH concentration (Fig. 4a) compared to the DOC concentration (Fig. 4b). The fit based on DOM-RSH was particularly improved at low DOM-RSH concentrations of ~2–50 nM, corresponding to ~50–150 μM DOC. The empirical relationship between MeHg VCF and DOM-RSH concentration was best fit by a power function (VCF = 0.67 × [DOM-RSH]$^{-0.4}$, $R^2$ = 0.96, $p$ < 0.001) instead of the more commonly used exponential relationship for VCF vs DOC[11,25].

The results support that the total DOM-RSH concentration is a primary controlling factor of MeHg bioavailability and cellular accumulation by phytoplankton. The significant and strong relationship between DOM-RSH and MeHg VCF corroborates the finding that other factors, like kinetic constraints, are of less importance in controlling MeHg uptake than speciation with DOM-RSH. The relationship further points to differences in DOM thiol concentrations as a likely principal factor behind the high, previously unexplained variability in MeHg VCF among different types of DOM at low DOC concentrations observed in previous studies[15,49]. Site-specific DOM-RSH concentrations alone are a much better predictor of MeHg VCF than DOC concentrations and should be quantified in experimental, modeling, and field-oriented research on MeHg speciation and bioaccumulation.

Based on current theory, two cases can be invoked to explain the power function relationship between MeHg VCFs and DOM-RSH concentrations (Fig. S5). The DOM-RSH concentration may control the equilibrium partitioning between MeHg bound to DOM-RSH and accumulated in plankton cells via reversible uptake mechanisms (case A). Alternatively, it may control the equilibrium distribution between MeHg(DOM-RS) and inorganic MeHg complexes (e.g. MeHgCl) in solution which are accumulated in cells with species-specific rates via quasi-irreversible mechanisms (case B). Case A presents MeHg cellular uptake as a thermodynamic equilibrium function between one MeHg species in solution (MeHg(DOM-RS)) and one associated with cells (MeHg(cell)$^{n+1}$) with a conditional constant $K_8$ (equation (S1d)). Rearranging the equilibrium expression of MeHg between cellular ligands and DOM-RS results in the MeHg VCF (i.e. [MeHg(cell)$^{n+1}$]/[MeHg(DOM-RS)]) being proportional to 1/[DOM-RSH] assuming the term $K_8$[H$^+$][cell] is constant under the experimental conditions (full equations in SI). With the limited data generated for this study, the empirical relationship between VCF and 1/[DOM-RSH] is not linear in the space as was expected (Fig. S5), thus the equilibrium model likely does not fully capture the complex process of MeHg uptake.

Case B accounts for a simplified, simultaneous uptake of the dominant organic and inorganic forms of MeHg in the experimental assays (MeHg(DOM-RS) and MeHgCl, respectively) based on their unique uptake rate constants and specific concentrations. Prior studies denote a faster uptake of MeHgCl due to passive diffusion across the cell membrane[22] while uptake of MeHg(DOM-RS) complexes are thought to be slowed by interactions at the cell surface prior to internalization[23]. The uptake rate constant of MeHg(DOM-RS) was estimated based on the highest DOM exposure concentration used in this study at 0.004 amol μm$^{-3}$ hr$^{-1}$ nM$^{-1}$. The MeHgCl uptake rate constant was set to fit the data, resulting in a value of 0.5 amol μm$^{-3}$ hr$^{-1}$ nM$^{-1}$; at the high end of published values (e.g., 0.175[12] and 0.5[49]). The Case B modeling results are plotted in Fig. S5 against the RSH concentration of the exposure medium. The model fit the

results well, but further mechanistic studies on cellular internalization of MeHg are required to better parameterize uptake as a direct function of MeHg speciation, cellular abundance, and cell size. Based on our modeling we do, however, suggest that bioaccumulation of MeHg into phytoplankton needs to be determined using a combined thermodynamic-kinetic model that considers water column MeHg speciation and the differences in the uptake rate constants for the different MeHg complexes.

## MeHg speciation and bioavailability in natural systems

Identifying and quantifying the DOM properties that dictate MeHg speciation and bioavailability, the core results of this study, provide a basis to substantially refine MeHg speciation and uptake models in coastal waters. We calculated the in situ MeHg speciation for the conditions at each site at the time of sampling (Table S4, SI Text Model 2). The modeling predicted that >99% of MeHg exists as MeHg(DOM-RS) complexes across the entire continuum despite that the $Cl^-$ concentration varied from 8 to 562 mM. At DOM-RSH concentrations between 5 nM (the lowest in our study) and 35 nM, the fraction of MeHg(DOM-RS) complexes varied in a narrow range between 99.09% and 99.99%, and was ≥99.99% when DOM-RSH was >35 nM (Fig. S6). The remaining MeHg fractions were dominantly MeHgCl and minorly MeHgOH. In marine systems, prior speciation models often predicted $Cl^-$ as the dominant MeHg-binding ligand. For instance, Zhong and Wang[18] proposed that MeHgCl would dominate over MeHg complexes with DOM in high salinity (300–500 mM $Cl^-$), low DOC (<100 μM) waters, conditions met in the open ocean. Using an updated MeHg(DOM-RS) average stability constant of 16.7 and RSH/DOC ratios from the shelf sites (Table 1), our modeling predicts that $Cl^-$ is the dominant binding ligand at 500 mM $Cl^-$ only when DOC is less than ~0.05 μM, conditions unlikely to be met in the global ocean[50].

The dominance of MeHg(DOM-RS) species does not negate the potential for significant MeHgCl uptake. Going back to case B described for the uptake experiment, the fast uptake of MeHgCl means that even a relatively small fraction may drive MeHg uptake in certain natural waters and this species should not be ignored. The unique uptake rates of MeHgCl and MeHg(DOM-RS) mean that the relative proportion of MeHg complexes in a system is significant for understanding its incorporation into cells. Based on the speciation model in Table S3, the molar ratio of the complexes MeHgCl and MeHg(DOM-RS) is related to $[Cl^-]$, 1/[DOM-RS], $K$, and pH. Our results show that the MeHgCl/MeHg(DOM-RS) molar ratio across the terrestrial to marine continuum was significantly, positively correlated with 1/[DOM-RSH] ($r^2 = 0.50$, $p = 0.001$), but not with $[Cl^-]$, log $K$, or pH. The small variability in MeHg speciation (including the magnitude of the MeHgCl fraction) was thus primarily driven by differences in the DOM-RS concentration among sites and not by $Cl^-$. This result highlights DOM-RSH concentration as the major control of MeHg availability for cellular uptake, regardless of the exact uptake mechanism.

Extrapolating our experimental results onto studies with conditions outside of what was examined here is associated with increased uncertainty, but is done to illustrate how expected gradients in DOM-RSH thiol concentrations across and within systems may clarify previous observations. For example, observed DOM-RSH concentrations in the North Pacific Ocean were lower compared to our offshore North Atlantic and nearshore Long Island Sound sites; 1–5 nM[51], ~5 nM, and 13–35 nM respectively. Seston MeHg BCFs were measured to be greatest in the Pacific (4.33–6.12 for the 0.2–5 μm fraction[52]) compared to offshore North Atlantic (2.67–4.41 for the same size fraction)[53] and Long Island Sound (2.55–3.25)[53], thus decreasing with increasing DOM-RSH concentrations. A similar approach can be extended to riverine systems. Luengen et al.[11] and Pickhardt and Fisher[21] both measured MeHg VCFs in laboratory experiments using water collected upstream of the San Francisco Bay. Pickhardt found a curious relationship where

uptake was higher in their high DOC site compared to their low DOC site: a situation not readily apparent in the Luengen study. A watershed analysis of the sites in both studies suggests that Pickhardt and Fisher's anomalous low DOC, low uptake site was located near a marsh system. Based on our results, it can be hypothesized that the marsh site had elevated DOM-RSH and therefore suppressed uptake to a greater extent than DOC concentrations alone would predict. This effect, however, remains to be tested for the specific sites.

There are additional factors that contribute to differences in MeHg BCF across systems such as phytoplankton abundance, community composition, cellular size distribution, and dissolved MeHg concentrations[12,25]. Further, laboratory studies have shown that the cellular uptake rate of MeHg-thiol complexes by phytoplankton[12,23] (and similarly the uptake of inorganic Hg-thiol complexes by bacteria[54,55] can differ depending on the chemical structure of the DOM-RSH compound (e.g. molecular weight and potential for chelation effects). More work is needed to clarify to what extent the molecular composition of thiols varies in natural DOM, and if this is a contributing controlling factor for MeHg bioavailability in the environment[56]. Our results highlight the significant improvement that quantifying DOM-RSH can have when parameterizing MeHg BCF results from natural systems instead of relying on DOC as the binding ligand proxy.

## Implications

This study provides thermodynamic and kinetic information on the interactions between MeHg and DOM extracted from a range of coastal locations, and a quantitative assessment of how DOM controls MeHg bioavailability to phytoplankton. The key results are conceptually visualized in Fig. 5. The figure illustrates how DOM-RSH concentration controls MeHg bioavailability (Fig. 5a), and how decreasing trends in DOM-RSH concentrations across the terrestrial to marine aquatic continuum (Fig. 5b) leads to higher MeHg availability and planktonic uptake in systems dominated by marine DOM relative to those impacted by more humic, terrestrial DOM (Fig. 5c). Decreasing DOM-RSH trends are driven by decreasing DOC concentrations and RSH/DOC ratios, the latter of which are impacted by the DOM source. A comparison of the results from field observations in terrestrial[19], estuarine, and/or marine aquatic systems[17,52,53] show higher MeHg concentrations in seston from systems with higher levels of marine DOM over terrestrial DOM. Such trends have also been observed in laboratory uptake experiments[18]. Where these previous studies invoke various principles in an effort to explain differences in uptake, such as higher aromaticity[19], larger molecular weight[17], and speciation with inorganic ligands[14,18] we can now more clearly link MeHg uptake to the DOM-RSH concentration. Specifically, our results, including the uptake experiment, suggest that changes in bioavailability are driven by MeHg speciation with DOM-RSH, which is not well predicted by local DOC concentrations.

Understanding how DOM controls MeHg bioavailability is critically important to predict MeHg exposure to high-trophic biota, including humans, now and under future scenarios[6]. Environmental change scenarios are expected to directly impact DOM loading to aquatic ecosystems, and therefore MeHg bioavailability[57]. For instance, eutrophication leads to the increased production of autochthonous DOM[58,59] which tends to produce DOM with low RSH/DOC. Allochthonous, terrestrial DOM inputs to aquatic systems are impacted by changes in watershed runoff due to e.g. climate change or land use and tend to deliver DOM with high RSH/DOC to coastal systems. Based on our results, increased proportions of fresh, marine DOM would lead to disproportionately smaller increases in local DOM-RSH than if terrestrially derived DOM increased in a system. The use of DOC as a proxy for MeHg-binding capacity would thus underestimate MeHg availability in algal blooms and overestimate it for increased terrestrial runoff.

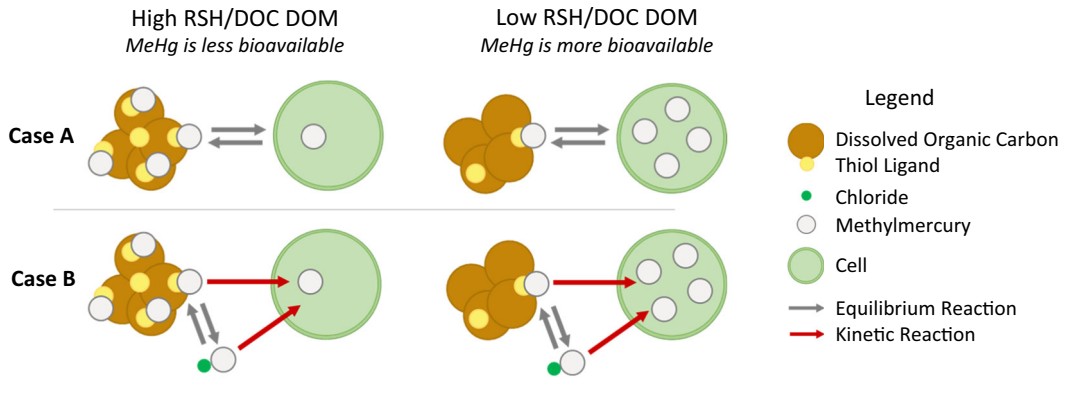

**a) DOM-RSH controls MeHg cellular uptake**

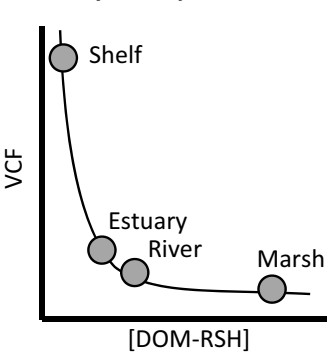

**Fig. 5 | Illustration of methylmercury bioavailability controlled by dissolved organic matter thiol concentrations.** Conceptual diagram depicting that dissolved organic matter's thiol ligand content (DOM-RSH), not carbon content (DOC), dictates methylmercury (MeHg) bioavailability to plankton cells across the terrestrial-marine aquatic continuum. **a** shows how the DOM-RSH concentration may control uptake by a direct equilibrium partitioning of MeHg between DOM-RSH and plankton cells (case A) and/or by an equilibrium between MeHg(DOM-RS) and MeHgCl complexes internalized by cells with species-specific rates (case B). **b** DOM-RSH (nM) and DOC (μM) concentrations across the terrestrial-marine aquatic continuum are shown using system-averaged bar plots (full dataset presented in Table 1). The numerical values above the bars are the system specific RSH/DOC (nM/μM) ratios. The substantial and systematic change in DOM-RSH concentrations across the continuum is driven by a decrease in both RSH/DOC ratios and DOC concentrations, resulting in a larger range of observed DOM-RSH compared to DOC. The implications of **a** and **b** are conceptualized in **c**. Namely, the bioavailability of MeHg (depicted as volume concentration factors (VCF) in plankton) increases in the order: marsh < river ≈ estuary < shelf because of the decreasing DOM-RSH concentration in water. Source data are provided as a Source Data file.

Our study highlights that any environmental process causing an increase in the molar concentration of DOM-RSH will decrease MeHg bioavailability. This effect is expected to be large at low to moderate nanomolar-level concentrations of DOM-RSH (≤ ~ 25 nM in our uptake experiment, Fig. 4) and minor at higher concentrations. The principal interactions between MeHg and DOM-RSH presented in the study provide a refined framework for how to consider dissolved MeHg speciation in MeHg uptake models. While more work is needed to better understand the mechanisms for uptake, more frequent DOM-RSH measurements in tandem with VCF or uptake experiments will lead to stronger empirical fits between dissolved MeHg and uptake across a range of sites. Together, this and future studies will become an important feature to refine models predicting MeHg incorporation in aquatic food webs for current and future Hg emission scenarios under various environmental change scenarios.

## Methods

### Sample collection and preparation

Water samples were collected using clean techniques by hand from shore at all but the shelf sites. The Gulf of Maine sample was collected by lowering a bucket over the side of a small boat, and the shelf break sample was collecting from 25 m depth using a Niskin bottle. DOM was extracted from 20–40 L (pending water color and expected DOC concentration) of each water sample within 24 hours of collection using the solid-phase extraction technique described by Dittmar et al.[26] taking care to load less than 10 L of sample or 2 mmol DOC per gram of sorbent. Briefly, the seawater was filtered to <0.2 μm using a pre-cleaned 0.45 μm Meissner cartridge filter followed by a 0.2 μm glass fiber capsule filter, and then acidified to pH 2 using trace metal grade hydrochloric acid before loading onto a clean modified benzene styrene polymer cartridge at a flow rate of <40 mL/min under the limitations set by Dittmar et al.[26]. The loaded cartridge was rinsed with 0.01 M hydrochloric acid and dried under Ar before eluting with methanol and acetone. The organic solvents were dried under nitrogen (N-EVAP 111) at 40 °C and the residue was freeze-dried.

### DOM characterization

The extracted DOM was characterized by total C, N, and S content, as well as S speciation and spectral indices. Dissolved organic C and total N were quantified using a Shimadzu TOC-V + TNM-1 analyzer at the Swedish University of Agricultural Sciences (SLU; Umeå, Sweden) after

acidification of the sample (Triplicate analytical percent relative standard deviation was 0.85 and 2.36 for DOC and total N, respectively). Total sulfur was determined via ICPMS[60]. Briefly, 1 mL of 10 M hydrochloric acid was added to a dry, weighed DOM sample followed by 3 mL of 12 M nitric acid. After a series of sonication and heating steps, 1 mL of 30% (w/w) hydrogen peroxide was slowly added to the solution. When the reaction terminated, the samples were heated again before diluting for analysis via ICPMS using magnesium sulfate for the standard curve. The CRM 129 (Hay Powder) was used to verify S recovery ($102.3 \pm 9.6\%$).

Between 24 and 40 µg C of freeze-dried DOM were reconstituted in 4 mL 0.1 M phosphate buffer (pH 7) to determine optical properties. Optical measurements were carried out with a Shimadzu UV 2600 (absorbance) and a Horiba AquaLog (fluorescence) with quartz cuvettes (10 mm path length). Phosphate buffer was used as the measurement blank. Absorbance was determined between 200 and 800 nm (1 nm increment) and fluorescence emission was determined between 250 and 820 nm (increment ~4.5 nm) at excitation wavelengths between 240 and 480 nm (3 nm increment). Further data processing was carried out with the drEEM toolbox (0.5.1[61]). Inner-filter effects were corrected using the absorbance-based method[62] and Rayleigh and Raman scatter were replaced with missing numbers and not interpolated. The raw fluorescence signals in arbitrary units were normalized to the area of the Raman peak at 350 nm. Humification index, fluorescence index, freshness index, and spectral slope ratio were determined with the drEEM toolbox as described elsewhere[63–66] (Fig. S7). Parallel factor analysis of the fluorescence (PARAFAC) data was carried out in conjunction with the N-way toolbox[67] (Fig. S8). All models were initialized with orthogonalized random values, constrained to nonnegative scores and loadings, and considered converged when the relative improvement in fit between iterations was smaller than $10^{-6}$. Between three and six components were considered. Ultimately, a four-component model that explained 99.9% of the dataset with a core consistency of 55% was splithalf-validated. It should be noted that the dataset very likely contained more components, but the sample size prevented the robust identification of further components. The four PARAFAC fluorescence components were named according to their emission peak position (C360, C390, C450, and C520). Due to the difficulty of quantitatively assessing the fluorescence intensities in solid-phase extracts, all component scores (i.e., the relative component abundances) were normalized to the sum of the fluorescence scores in each sample.

Sulfur K-edge XANES data were collected on beamline 4B7A equipped with a Si (111) double-crystal monochromator at the mid-energy X-ray station in the Beijing Synchrotron Radiation Facilities, China. The storage ring was 2.2 GeV and ring current 100 mA. The dry DOM samples were rubbed onto sulfur-free tape in a thin layer for the analysis. The sample was then mounted to the sample cell and flushed with He. Measurements were performed under high vacuum ($10^{-6}$ to $10^{-8}$ mbar) at ambient temperature and the incident X-ray energy range scanned was 2462 to 2500 eV with a step size of 0.2 eV. The reference compounds included sodium sulfate ($Na_2SO_4$), sodium sulfite ($Na_2SO_3$), elemental sulfur ($S^0$), cysteine (-SH), methionine ($-S=$), sodium methane sulfonate ($CH_3SO_3Na$), and iron sulfide (FeS). All chemicals were analysis grade purchased from Sigma-Aldrich except for FeS which was purchased from Alfa Aeser. The resulting sample spectra were processed following the method of Yekta et al.[68] (Fig. S9). First, a polynomial pre-edge function was subtracted from the spectrum. The data were then normalized between 2460 and 2490 ev using the software WinXAS97. Microsoft Excel was used to deconvolute the corrected spectrum into sulfur pseudo-components (representing typical reference compounds) using a least-square fitting procedure. The results are summarized in Table S5.

The total concentration of DOM-RSH was determined on the DOM extract by titration with monobromo(trimethylammonio) bimane bromide (qBBr) and quantification of excess qBBr by LC-ESIMS/MS[33].

The PCA plots were generated in R Studio version 1.4.1103 using the prcomp{stats} function (stats version 4.0.3)[69]. The data were centered and scaled using arguments within the function. The S speciation data in Fig. 2c, d included seven S XANES pseudo-components as percentages of total sulfur, and the independently quantified total DOM-RSH concentration as a percent of measured total S. All boxplots were also generated in R Studio. The top and bottom edge of each box correspond to the first and third data quartiles, while the center line represents the data mean. The whiskers extend to the furthest data points not considered outliers, defined as values beyond 1.5 times the interquartile range.

## LC-ICPMS methods

The MeHg(Nacpen) complex was measured using LC-ICPMS (Perkin Elmer Elan DRC) (adapted from Liem-Nguyen et al.[70]). The LC column used was a Phenomenex Kinetex Biphenyl column (150 mm × 3 mm × 5 mm) with a 4 × 3 mm Phenomenex guard column. The column oven temperature was set to 25°C. The mobile phase was 7% (v/v) 1-propanol in 0.5 mM phosphate buffer. The liquid flow rate was set to 0.4 mL/min with a 0.1 mL/min post column flow rate of a solution containing thallium (10 ng/mL) to assess the sensitivity of the ICPMS day to day and over the course of the experiment. A MiraMist nebulizer and a cyclone spray chamber cooled to +4°C was used. The nebulizer and auxiliary gas flow rates were set to 0.6 and 1.2 L/min, respectively. External calibration was used for calculating the sample concentrations using a range of MeHg(Nacpen) standard concentrations in 0.5 mM phosphate buffer, for which the concentration was verified using a direct mercury analyzer. The LC-ICPMS peaks were integrated manually using OriginPro 9 software and normalized to the thallium signal prior to calculating the MeHg(Nacpen) concentrations for both the standards and the samples. Further, the MeHg(Nacpen) concentration of the MeHg(Nacpen) standards used for competing ligand exchange experiments was quantified for each individual sample batch by ICPMS with an average recovery of 116%. The recovery was accounted for in the log $K$ calculations (described below). More information on blanks and recoveries is detailed in the 'Competing Ligand Exchange Experiments' sections below.

## Gas chromatography (GC)-ICPMS methods

Subsamples for quantification of total MeHg concentration were collected after the completion of the LC-ICPMS analysis and immediately preserved by acidification to 0.5% sulfuric acid and stored cold. Total MeHg concentrations were determined using isotope dilution analysis and measured using an Agilent 7890 GC (HP-1 column, 30 m, 0.320 mm wide bore, 1.00 µm film) coupled to an Agilent 7700x ICPMS. An Me$^{201}$Hg internal standard was added to the samples and allowed to equilibrate for 24 hours before digesting with 1% (v/v) sulfuric acid for another >12 hours. 4.5 M acetate buffer was subsequently used to adjust the sample pH and the MeHg was ethylated using sodium tetra ethyl borate and extracted into toluene. Once all reagents were added, the sample was mixed for at least 1 hour prior to transferring to GC vials for analysis. The toluene samples were injected to the instrument as a 1 µL split sample injection. The injection temperature was 150 °C and the GC separation was done isothermally at 80 °C. The ICP radio frequency power was 1350 W and the monitored isotopes were $^{200}$Hg, $^{201}$Hg, $^{202}$Hg, $^{204}$Hg with a dwell time of 30 ms. The total acquisition time was 252 s, and the derivatized MeHg retention time was 196 s. The same MeHg(Nacpen) standards used to determine the LC-ICPMS recovery were also analyzed for total MeHg by GC-ICMPS with an average recovery of 110%. More information on the recoveries are detailed in the 'Competing Ligand Exchange Experiments' sections below.

## Competing ligand exchange experiments and determination of log $K$ for MeHg(DOM-RS)

Standard competing ligand exchange techniques were used in the overall design of the experiment[71], but were refined by the direct quantification of the MeHg complex with the competing ligand, i.e., MeHg(Nacpen), using LC-ICPMS[34] as described above. Solutions of DOM, MeHg, and Nacpen ($(CH_3)_2C(SH)CH(NHCOCH_3)CO_2H$) were made under anoxic conditions. Specifically, a known mass of the DOM sample was reconstituted in $N_2$ sparged 0.5 mM phosphate buffer such that the final DOM concentration was 500 mg/L and the pH 7.5. All solutions were kept in the dark and were rotated overnight prior to filtering to <0.2 μm using syringe filters. Methylmercury (~100 nM MeHg as a Cl species) and Nacpen were added simultaneously to the DOM (~85 mg/L DOM) solutions, mixed, and left to rotate for 24 hours prior to analysis. The MeHg and DOM concentrations were kept constant across the experimental samples while Nacpen was added at seven different concentrations between ~0.05 and 50 μM with slight variations between DOM types, and most samples were made and analyzed in duplicate (~14 total experimental samples per DOM).

The analytical standard deviation between experimental duplicates was on average 6 nM for the LC-ICPMS measurements and 3.8 nM for the GC-ICPMS measurements. The controls for the experiment included a matrix blank (buffer + 2 μM Nacpen; average from all runs = $1.3 \pm 2.5$ nM MeHg(Nacpen) measured by LC-ICPMS and $4.9 \pm 6.7$ nM total MeHg measured by GC-ICPMS), a matrix blank with 82 mg/L DOM added (average from all runs = $5.8 \pm 2.5$ nM MeHg(Nacpen) and $6.6 \pm 6.8$ nM total MeHg), and a DOM blank with 82 mg/L DOM and 100 nM MeHg added (average from all runs = $40 \pm 57$ nM MeHg(Nacpen) and $93 \pm 11$ nM total MeHg). In theory, no MeHg(Nacpen) should have been detected in the DOM blank sample. However, we found that memory effects on the LC column occasionally caused elevated sample concentrations above what was expected based on GC measurements. Therefore, the DOM blank was subtracted from the MeHg(Nacpen) values when elevated in respect to the GC-ICPMS measurement, which included ten DOM experiments (blank range = 1.2–62 nM, mean = 15.9 nM). A new stock solution of Nacpen was made for each competing ligand exchange experiment. Therefore, a standard with MeHg in excess of Nacpen was used to verify the Nacpen concentration of the stock solution and used to correct the final Nacpen concentrations in the speciation calculations. A sample with Nacpen in excess of MeHg was also run to verify the MeHg concentration of the working solution, which was thawed from a frozen stock before each use (average from all runs = $105 \pm 18.7$ nM MeHg(Nacpen) after blank correction and $111 \pm 11.1$ nM total MeHg). Expected concentration was 100 nM MeHg(Nacpen) and total MeHg.

To assure MeHg was stable over the timeframe of the experiment, preliminary experiments were carried out that showed insignificant degradation over the period of seven days. Effectively all MeHg was assumed to be complexed with Nacpen or DOM-RS⁻ based on speciation modeling results (using the WinSGW software). The concentration of MeHg(DOM-RS) complexes was thus calculated as the difference between the total MeHg concentration determined independently with GC-ICPMS, and the MeHg(Nacpen) concentration determined by LC-ICPMS. The concentration of free DOM-RSH ligands was calculated as the total DOM-RSH concentration, determined by the mass of DOM added to the reaction and the DOM-RSH/g DOM quantified by the qBBr-titration, minus the concentration of MeHg(DOM-RS) complexes.

The overall competing ligand exchange reaction is described by:

$$\text{MeHg(Nacpen)} + \text{DOM} - \text{RS}^- = \text{MeHg(DOM} - \text{RS)} \\ + \text{Nacpen}^- ; K_1 (= K_2/K_3) \tag{1}$$

based on the four individual reactions:

$$\text{MeHg}^+ + \text{DOM} - \text{RS}^- = \text{MeHg(DOM} - \text{RS)} ; K_2 \\ = \text{determined in the competing ligand exchange experiment} \tag{2}$$

$$\text{MeHg}^+ + \text{Nacpen}^- = \text{MeHg(Nacpen)} ; K_3 = 10^{16.76} \tag{3}$$

$$\text{Nacpen} - \text{H} = \text{Nacpen}^- + \text{H}^+ ; K_4 = 10^{-9.6} \tag{4}$$

$$\text{DOM} - \text{RSH} = \text{DOM} - \text{RS}^- + \text{H}^+ ; K_5 = 10^{-9.6} \tag{5}$$

Reactions 4 and 5 describe the acid dissociation reaction of the thiol groups in Nacpen and DOM, respectively, and the numerical values of the corresponding constants were taken from Liem-Nguyen et al.[70]. The concentrations of MeHg species with $Cl^-$, $PO_4^{3-}$ and $OH^-$ were negligible in these experimental systems but the species were nevertheless included in the complete speciation model (Table S3, Eq. (2)-(12)). For each individual DOM sample, the chemical speciation of MeHg was calculated for the seven Nacpen additions (0.05–20 μM). The model was based on the input variables: total concentrations of MeHg, Nacpen, DOM-RSH, $Cl^-$ and $PO_4^{3-}$, pH, and the known stability constants in Table S3 (SI Text Model 1). The stability constant for MeHg(DOM-RS), i.e., log $K_2$, was varied between 15 and 17. For each log $K_2$ tested, a root mean square error (RMSE) was calculated between the MeHg(Nacpen) concentrations determined by the speciation modeling and by the direct LC-ICPMS measurements. The RMSE's were plotted versus the log $K_2$ values, fit against a quadratic formula, and solved for the log $K_2$ achieving the lowest error. The determined values of $K_2$ are thus linked to the constants $K_3$–$K_5$ and the competing ligand exchange experimental uncertainty is mainly determined by the uncertainty in the measured MeHg(Nacpen) concentration, which typically was up to 20%. We estimated the experimental uncertainty in log $K_2$ by numerically varying in the calculation scheme the MeHg(Nacpen) concentration within 20% of the measured value. This resulted in an average uncertainty in log $K_2$ of ±0.19, which is a very similar uncertainty as in previous studies determining log $K$ for inorganic Hg complexes with thiol compounds using competing ligand exchange-based approaches[34,35,72].

## Kinetics of the MeHg ligand exchange reaction

The kinetic ligand exchange experiment was set-up in a similar way as the competing ligand exchange experiment, but one isotope of MeHg was pre-equilibrated to the DOM ($Me^{204}Hg$) and another was pre-equilibrated to Nacpen ($Me^{200}Hg$) for four days prior to mixing and analyzing. The MeHg isotope standards were added at a 2:1 ratio in the final mixture (100 nM $Me^{204}Hg$, 200 nM $Me^{200}Hg$). Five DOM samples were tested; PAW (riverine), NB (estuarine), BI and SI (marsh), and SB (shelf break). Three different competing ligand (Nacpen) concentrations were tested such that the ratio of DOM-RSH to Nacpen was approximately 1:0.1, 1:1, and 1:10. Immediately after mixing, a sample was removed and the MeHg(Nacpen) concentration for both $Me^{204}Hg$ and $Me^{200}Hg$ was determined via LC-ICPMS. Samples were then removed from the anaerobic chamber and analyzed every ten minutes (the length of one analysis run) for the first hour of the reaction, and then periodically over the course of the following five days to ensure the samples had reached equilibrium. The $^{204}Hg/^{200}Hg$ isotope ratio for MeHg(Nacpen) ($R_{\text{MeHg(Nacpen)}}$, determined by LC-ICPMS) was divided by the $^{204}Hg/^{200}Hg$ isotope ratio for total MeHg in the system ($R_{\text{total MeHg}}$, determined by GC-ICPMS) and plotted versus time (Fig. S2). At equilibrium, the two isotope ratios equal, i.e., $R_{\text{MeHg(Nacpen)}}/R_{\text{total MeHg}} = 1.0$. The presence of a "non-exchangeable pool" is indicated by a systematic deviation from 1.0 in the $R_{\text{MeHg(Nacpen)}}/R_{\text{total MeHg}}$

ratio at steady-state. Such deviation was less than 10% for all the tested DOM samples (Fig. S2). Pseudo-first order rate constants for MeHg ligand exchange between Nacpen and DOC-RSH was calculated (R software) based on the measured Me$^{204}$Hg(Nacpen) mole fraction of total Me$^{204}$Hg following Skrobonja et al.[23]:

$$f(Me^{204}Hg(Nacpen)) = \frac{k_f}{k_r + k_f} \times \left(1 - e^{-(k_r + k_f)t}\right) \qquad (6)$$

where $k_f$ is the rate constant for the forward reaction (Me$^{204}$Hg(DOM-RS) → Me$^{204}$Hg(Nacpen)) and $k_r$ for the reverse reaction (Me$^{204}$Hg(Nacpen) → Me$^{204}$Hg(DOM-RS)). The turnover (($k_f + k_r$)$^{-1}$) was then calculated from the two constants.

**MeHg cellular uptake experiment**

Artificial seawater (psu = 35; 19.52 g/L Cl$^-$, 11.94 g/L Na$^+$, 5.27 g/L SO$_4^{2-}$, 1.33 g/L Mg$^{2+}$, 0.36 g/L Ca$^{2+}$, 0.42 g/L K$^+$, 0.36 g/L HCO$_3^-$; modified from[73]) was made using acid cleaned glassware and pre-combusted salts to maintain a low DOC contamination. DOM extract from sites SI or OR was added to the artificial seawater to final DOC concentrations of 50, 150, and 500 μM, equating to 17–170 and 2.2–22 nM DOM-RSH for SI and OR, respectively. MeHg was added to the DOM solutions to a final concentration of 10 pM, allowed to equilibrate overnight, and then split into triplicate 200 mL aliquots for each DOM type and DOC concentration. After the equilibration period, 2.4 × 10$^7$ cells of *Thalassiosira pseudonana* were added to each 200 mL flask (1.2 × 10$^5$ cells/mL), determined by counting the cell density of the stock culture with a Coulter Multisizer IIe, and 30 mLs of the diluted culture was immediately filtered to obtain the initial MeHg particulate and dissolved concentrations. The cells were incubated under full spectrum lights for 4 hours, then 30 mLs from each flask were filtered. The filters were preserved by freezing while the filtrate was preserved with 1% sulfuric acid. The final cell count of the incubated culture was determined using a Coulter Multisizer IIe. The MeHg samples were processed and analyzed following standard techniques[74,75]. Briefly, the filters were digested in 4.5 N nitric acid overnight. A digest aliquot was neutralized with potassium hydroxide and acetate buffer and ethylated using sodium tetraethylborate. The sample was then measured on a Tekran 2700 Automated Methylmercury Analysis System by gas chromatography and cold vapor atomic fluorescence detection with calibration against a standard curve (Alfa Aesar CAS: 115–09–3, LOT: 1791821). The MeHg concentration was above the analytical detection limit of 0.004 ng/L (range = 0.018–0.532 ng/L), and analytical duplicates had RSDs of 15 ± 6.0% (*n* = 4) for all the particle digests. Dissolved MeHg samples were analyzed similarly except 2.5% L-ascorbic acid was added when the sample was neutralized[75] (Alfa Aesar CAS: 115–09–3, LOT: 1791821). All samples were above the analytical detection limit (0.35–1.05 ng/L) but were not run in duplicate. Treatment replicates, however, had an average RSD of 11 ± 5.3% and spike recoveries of 97 ± 24%. The suspended particulate MeHg concentrations were converted to nmol MeHg/cell volume for the VCF calculation. To do so, a cell diameter of 8 μm was used (113.1 μm$^3$) based on the average cell size from the Coulter counter, and the initial MeHg cellular concentrations were subtracted from the final cellular concentrations.

## Data availability

All data that support the findings of this study are provided in the paper and the Supplementary Information. Further information regarding the data processing to achieve the datasets presented within the source data file is available from the corresponding authors upon request. Correspondence and requests for materials should be addressed to E.S. or E.B. Source data are provided with this paper.

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

## Acknowledgements

This work was financially supported by the US National Science Foundation (GRFP: DGE-1747453), the Swedish Research Council (2016-06459), and the Kempe Foundations (SMK-1243, SMK-2745). We thank Khoa Huynh for carrying out the total DOM-RSH measurements, Dr. Yu Song at the Swedish University of Agricultural Sciences and Dr. Chenyan Ma at Beijing Synchrotron Radiation Facility (Beamline 4B7A) for assistance with the sulfur K-edge XANES spectroscopy measurements. We also thank Patricia Meyer at the University of Connecticut for her assistance in the phytoplankton uptake experiment.

## Author contributions

E.S. and E.B. conceived and designed the research. E.S. performed most of the experiments. V.L.-N. contributed to the mass spectrometry measurements and U.W. performed the fluorescence spectroscopy measurements and data analysis. E.S., E.B., Z.B., R.M. and U.S. performed data analyses. E.S. and E.B. wrote the original manuscript draft and all authors contributed to reviewing and editing the manuscript.

## Funding

## Competing interests

The authors declare no competing interests.
