## [Peer Review File · Nature Communications]

Dissolved organic matter thiol concentrations determine methylmercury bioavailability across the terrestrial-marine aquatic continuumReviewer #1 (Remarks to the Author):

This paper addresses a long-standing question in the mercury community about why the effects of DOM on mercury uptake by phytoplankton are so variable. The authors use a variety of techniques to demonstrate that the amount of DOM-associated thiol groups is really key to predicting algal uptake of MeHg. In demonstrating this point, the authors also ruled out a couple of other factors that researchers have considered potentially important (e.g., Cl⁻ concentrations and kinetic control of MeHg partitioning). As such, the paper is an important contribution to the literature. The conceptual model diagrammed in Fig. 5 is also a very nice mechanistic contribution to the field, showing how both DOC concentrations and thiol groups associated with the DOM vary across a continuum of sites from marsh to river to estuaries to continental shelves.

However, the ability of this paper to achieve its full impact is hampered by how difficult it is to read. The technical content and methodology are outstanding, but much, much more attention is needed to present the material in a digestible format to the reader. This paper will be of wide interest to the mercury community, not just those who are experts in Hg/DOM interactions so the authors should spell out some of the implications to the reader, including giving an occasional background sentence. In addition to a little more explanation, a fair bit of work is needed to improve the writing, such as grouping material into paragraphs with clear topic sentences. The table and figures also need improvements to make them more understandable. Some of the text also seems like it should be moved from the results to the discussion. To this end, many of the comments below are designed to address the readability of the paper or to point out places where the implications should be discussed in more detail.

Abstract

Line 23: Awkward punctuation with the "e.g." Adding a comma and parentheses would help, as in ". . . in aquatic food webs (e.g., following various Hg emission and environmental change scenarios). But better yet combine into a single sentence, ". . . we formulate a generalizable MeHg bioavailability model to predict MeHg incorporation in aquatic food webs under various emission and environmental change scenarios."

Try to work some of the major numbers from the paper into the abstract. The finding on line 141 that the DOM-RSH (nM) thiols varied by a factor of 40 while the DOC only varied by a factor of 5 was a key one for me, but is not really reflected in the abstract.

The paper also emphasized that Cl⁻ speciation is not controlling MeHg bioavailability, but this finding is not emphasized in the abstract. Instead, the abstract says that salinity ranged from 0.5 to 36 ppt (check the units here). Stating the salinity in the abstract makes it sound like salinity is important, when from my reading of the paper, it is really about the DOM composition.

Work on summarizing the results, not just what was studied. For example, line 18 states, "We show that the concentration of DOM-associated thiol functional groups (DOM-RSH) dictates MeHg speciation and bioavailability across the terrestrial to marine aquatic continuum." It would be better if you gave the direction of the relationship (and maybe even the magnitude), as in, "As the RSH-DOM ratio decreases from terrestrial to marine systems, there is a resulting decrease in MeHg bioavailability to phytoplankton."

**To help illustrate how the abstract could be improved, see the classic piece by Kenneth Landes, *A Scrutiny of the Abstract*. I'm including the link here in case it is of use to graduate students/post docs involved with this project:
https://www.caam.rice.edu/~symes/CAAM600/abstract_scrutiny.pdf**

Introduction

The introduction is very clearly framed and succinctly summarizes the existing literature.

Line 27. Work on succinctness. For example, "It is estimated that anthropogenic Hg emissions have caused a three-fold increase in Hg content in surface marine waters compared to pre-industrial conditions . . ." could become, "Anthropogenic Hg emission have caused a three-fold increase . . ."

Results

Line 86. Remove the comma after et al., in ". . . Dittmar et al., (2008) and . . ."

Break up the long paragraph that begins on line 80. Lines 80 – 89 are a brief summary of methods. Start the results of the PCA analysis in a new paragraph and improve the topic sentence on line 90 so that it is clear that you will have a paragraph just on the spectral data. Use a different paragraph for the S speciation data.

Lines 90 – 96. Were there two separate PCA analyses done here? One on the spectral data and one on the S speciation? Some of the language in the text is confusing. For example, line 95 states, "The second PC described 27% of the data variability and discriminated fluorescent DOM by origin (C390 bacterially derived material, C520 terrestrially dominated material)." I think this is the first PCA analysis, but the second PC, but I had to closely study Fig. 2 to check this. If indeed there are two separate PCA analyses done here, describing each in its own paragraph would help considerably. You may even want to consider subheaders, such as paragraph header.

OK, as I'm reading along to line 103, I am seeing that there are two PCAs here. Not introducing the second PCA on line 92 would help. Wait and introduce the second PCA and Figure 2c and d on line 103 when you give the results for this PCA.

Line 112 – the huge variability between the estuarine sites is really interesting here and the explanation about mixing between both terrestrial and marine algal DOM makes sense.

Lines 146 – 168 – break up this long paragraph, perhaps by starting a new summary paragraph on line 161, with "Combined, these results. . ."

Line 167. Correct punctuation is ". . . experiments are, however, not straightforward. . ."

Line 201. The two isotopes were a nice approach to determine that the non-exchangeable pool of MeHg was very small.

Line 244. Can you put the reference to the relevant figure or table at the end of this sentence?

Line 263. Ah, it is so nice to see all of the variability in MeHg VCFs explained in terms of DOM-RSH.

Line 270. Oh no, it is hard to even begin to read this giant paragraph. Use paragraphs of 3 – 7 sentences to help your reader! Consider paragraph breaks on line 277 with the thermodynamic modeling. Make line 285 saying that MeHg speciation is not driven by Cl⁻ concentration the topic sentence of a paragraph on your thermodynamic results. The importance of Cl⁻ has been brought up enough in the literature that this result should be easy to find in the topic sentence. Can you move some of this material into the discussion? Some of the interpretation, such as that the role of Cl⁻ in controlling MeHg speciation has been overstated, sounds like it is more discussion than results.

Line 297. I can't wait to see your next study, where you include the open ocean. But again, I think this material is discussion, not results.

Line 309. Remember to offset "however" with commas.

The extensive use of acronyms in this manuscript don't help with readability. For example, N-Acetyl-Penicillamine is abbreviated Nacpen. Even though this acronym is used a lot during the manuscript, it was not one that I knew so it complicated the reading. Completing ligand exchange is CLE. While some of these acronyms (e.g., DOM, BCF, etc.) are quite common in the field, I had to repeatedly refer back to others. Could you systematically go through this manuscript and try to remove a couple of the less common and/or less frequently used acronyms so that the paper reads less like alphabet soup?

Readability is also hampered by some of the notation. For example, the paper refers both to the RSH/DOC ratio and also to the DOM-RSH, which is not a ratio, but is the concentration of DOM-associated thiol groups (line 19). Is there some clearer way to display DOM-RSH so the poor reader is not getting mixed up between the ratios and the concentration of thiol groups? Maybe just call it thiol concentration?

Discussion

Line 319. It should be ". . . how DOM controls MeHg bioavailability. . ."

Line 319. I think you are missing the word "of" and the tense is wrong. It should be ". . . a quantitative assessment of how DOM controls MeHg bioavailability. . ."

Lines 344 – 368. Can you separate out your discussion of how MeHg is taken up by phytoplankton cells and implications for Fig. 4 from your discussion of Fig. 5.

Figure 5 is a very interesting figure and nicely summarizes the paper, but is only mentioned briefly on lines 345 and 351. For such an important visual summary, this figure merits more play. Discuss it in its own paragraph. Also discuss it in light of the literature. For example, do field studies support the idea that shelf systems have the highest MeHg in algal cells? Does this translate up the food chain? And how does it tie in with the common observation that oligotrophic lakes often have very high MeHg concentrations in fish?

Line 374. Spell out what you mean by "potentially opposite impacts on MeHg formation and bioaccumulation." Which anthropogenic alteration is likely to have which effect?

Line 378. Are you anticipating an increase in DOC overall or a change in the RSH/DOC ratio? Would changing the concentration of both RSH and DOC without changing the ratio change VCFs?

Line 380. How would an algal bloom change the RSH/DOC ratio? Why would terrestrial runoff increase RSH/DOC ratio?

The results of the PCA analysis are never mentioned in the discussion. How does the PCA analysis relate to the material in the discussion on MeHg bioavailability to phytoplankton?

Overall, spend some more time on the implications of the importance of the DOM-associated thiols. I think what I'm hearing is that measuring DOC is not really sufficient to predict bioavailability to phytoplankton. Instead, XANES is needed to quantify the number of thiol groups associated with the DOM. But most researchers don't have XANES! So what would you recommend for researchers seeking to study global environmental change and for monitoring programs that still use DOC?

Methods

Line 418 sounds like it is statistical methods – separate it out into a new paragraph.

Check punctuation on line 439.

Lines 479- 515. Break up this long paragraph.

Line 499. What was the general magnitude of the DOM blank that had to be subtracted?

Line 564. Salinity units of ppt are not appropriate -see comments under Table 1.

Table 1.

Salinity in ppt is a rather old notation and not consistent with the use of a salinometer to measure it. When measured with a meter, salinity is measured as conductivity relative to a standard, which makes it unitless. Those who hate unitless numbers use practical salinity units (psu). In contrast, latitude and longitude do have units, which are not currently shown in the table.

Figure 1

Please add a few place names to the map, to orient people who are not very familiar with the East Coast of the United States. Also consider the legibility of this map when printed in black and white or viewed by people who are color blind. In addition to using colors for habitat type, use different symbols (e.g., triangles, squares, etc.) to distinguish between these habitats.

Figure 2

Please make the legend and axes and site names larger. I can hardly read them, even blown up 150% and when they are published in the journal, they are going to be even smaller.

Label some of the individual panels, such as with Fluorescence spectral data and the other with S speciation. The current labeling just says, "variables – PCA" which is not particularly helpful. Improve this heading. Write out the x-axis labels in panels b and d. I'm assuming that Dim1 is dimension 1? You are really making the reader work hard to interpret these panels. Write out axes labels and use the text and caption to help the reader interpret the figure. The addition of the oxidation states on panel d definitely helps, but add a sentence about the dimensions, particularly linking the dimensions, the loadings, and the contributions. I do not understand the scale of the "contributions" on panels b and d. I am used to thinking about component loadings ≥ 0.6 or ≤ -0.06 as being important but not really sure where the info for "contrib" came from, and I'm not getting that from the text, either.

Figure 3

Looks great. Very interesting progression in DOM thiol concentrations in panel 3a.

Figure 4

Put the name of the diatom species in the caption. Does this relationship still hold even if the point in the top-right corner of panel a is removed? Which points are which? It would be nice to see which points in figure 4a correspond to the points in 4b. Since there are so few points, coding them by using different symbols would be reasonable.

Figure 5

Would it make sense to show the RSH/DOC ratio to illustrate how it changes across the continuum of sites? The little squares that link from the bottom right or the bottom left or the middle of the bar charts to the cartoons are weird and not adding anything.

Reviewer #2 (Remarks to the Author):

Reviewer #2 Attachment on the following page

The manuscript is entitled "Dissolved organic matter thiol concentrations determine methylmercury bioavailability across the terrestrial-marine aquatic continuum"

General comments:

This is a manuscript which deals with the role of the DOM-Thiol groups for the complexation and bioavailability of MeHg across different aquatic systems. The authors determined the thermodynamic and kinetic interactions between MeHg and DOM-Thiol groups extracted from varied aquatic sites. The results are quite interesting to me to see such a good relationship between RSH concentrations and VCF MeHg in phytoplankton (Anyway, is there any relationship between them if we removed only one data of shelf sea in Fig. 4a). Additionally, the observation that K doesn't change too much is pretty valuable. However, one big gap is that the DOM they are studying is high molecular weight...so, this is only a part of the story. The strategy of the sampling also lacks the shelf data although the trends with that point make sense. Nature Comm is short like Nature. This seems like a pretty long paper, full of small details. I think it's well done and pretty important. I indeed learned a lot from this paper related to potential mechanisms How MeHg gets into the cell of phytoplankton.

Some minor comments:

=====

Line 118, "the chemical speciation and bioavailability of dissolved MeHg..." This is kind of a hypothesis. To be sure, this paper helps support this hypothesis, but this statement sounds too categorical.

Line 122 "DOM binding capacity" is a little too ambiguous a term in this context. Since it is DOC normalized, consider using something like "DOM-specific binding capacity" or something.

Line 129-132, "binding affinity..." citation for this method?

Line 133, how do these results compare to total Hg? Wouldn't we expect that their concentrations would be similar?

Line 199, the ambient ligands come out looking like Nacpen...this is what happens using CLE. Any comments on analytical "windows" that are created by this approach?

Line 224, complexation for MeHg sounds faster than for Hg(II)...any comments?

Line 261, "The results demonstrate that the total DOM-RSH..." The word "demonstrate" is a little too strong here. Change to "the results are consistent with the hypothesis that..."

Line 394, talk about the limitations of Dittmar approach...this is mostly high molecular weight material. By ignoring LMW thiols, the importance of complexation is underestimated a bit. But, LMW thiols have been reported to be especially bioavailable...so, the MW/LW distinction might be important to discuss.

Line 463, citation for this method?

Line 522, citation for the K6 value?

=====

Author Response to Reviewer Comments

We thank the reviewers for their positive overall assessment of our manuscript and for their valuable specific remarks and constructive criticism. We have carefully revised the manuscript based on the specific comments by the reviewers.

Both reviewers made a remark regarding the data point with highest VCF value in figure 4. It turned out that this data point was incorrect due to a calculation error. We are very grateful for the reviewers' remarks and we have gone through the entire data set of the manuscript to ensure there are no further errors. The rectification of the incorrect VCF data point does not affect the key findings of the manuscript, i.e. that DOM associated thiols control MeHg bioavailability and is a much stronger predictor for MeHg VCF than DOC is. Revision of the VCF data set, however, affects the theoretical interpretation of the observed relationship between DOM-RSH concentration and VCF. In the revised manuscript, we present one thermodynamic and one kinetic model (Lines 280-308 and Supporting Information Figure S5) describing how DOM-RSH control MeHg uptake and we note that the kinetic model fits our experimental results the best. We have further revised the layout of Figure 4 by plotting VCF versus the concentration of DOM-RSH (Fig. 4a) or of DOC (Fig. 4b) instead of plotting VCF versus 1/DOM-RSH or 1/DOC as in the original manuscript. We have also revised the relevant text in the Discussion and Abstract to harmonize with the revised Figure 4 and MeHg uptake models.

Additional major revisions made to the manuscript in response to reviewer comments are to merge the Results and Discussions into a joint section and to revise Figure 5 to more clearly communicate the three core messages of the manuscript regarding the role of DOM-RSH in controlling MeHg bioavailability.

In our point-by-point responses below our specified line numbers refer to the revised manuscript without track-changes indicated.

Reviewer #1 (Remarks to the Author):

This paper addresses a long-standing question in the mercury community about why the effects of DOM on mercury uptake by phytoplankton are so variable. The authors use a variety of techniques to demonstrate that the amount of DOM-associated thiol groups is really key to predicting algal uptake of MeHg. In demonstrating this point, the authors also ruled out a couple of other factors that researchers have considered potentially important (e.g., Cl⁻ concentrations and kinetic control of MeHg partitioning). As such, the paper is an important contribution to the literature. The conceptual model diagramed in Fig. 5 is also a very nice mechanistic contribution to the field, showing how both DOC concentrations and thiol groups associated with the DOM vary across a continuum of sites from marsh to river to estuaries to continental shelves.

However, the ability of this paper to achieve its full impact is hampered by how difficult it is to read. The technical content and methodology are outstanding, but much, much more attention is needed to present the material in a digestible format to the reader. This paper will be of wide interest to the mercury community, not just those who are experts in Hg/DOM interactions so the authors should spell out some of the implications to the reader, including giving an occasional background sentence. In addition to a little more explanation, a fair bit of work is needed to improve the writing, such as grouping material into paragraphs with clear topic sentences. The table and figures also need improvements to make them more understandable. Some of the text

also seems like it should be moved from the results to the discussion. To this end, many of the comments below are designed to address the readability of the paper or to point out places where the implications should be discussed in more detail.

Response:

We thank the reviewer for the overall positive feedback, and valuable remarks. As described in detail in our below responses to specific remarks we have made substantial revisions of the text to improve the readability of the manuscript, including to restructure the grouping of material, length of paragraphs and clarity and placement of topical and background sentences. We have further revised most of the manuscript figures for improved clarity. We also suggest a rearrangement of the text to a joint Results and Discussion section.

Abstract

Line 23: Awkward punctuation with the “e.g.” Adding a comma and parentheses would help, as in “. . . in aquatic food webs (e.g., following various Hg emission and environmental change scenarios). But better yet combine into a single sentence, “. . . we formulate a generalizable MeHg bioavailability model to predict MeHg incorporation in aquatic food webs under various emission and environmental change scenarios.”

Response:

We agree with the reviewer’s comments on the abstract and we have made substantial revisions of the text (see also responses to the remarks below). Regarding this specific remark, we have revised the text as follows:

Line 24:

“Our results strongly suggest that measuring DOM-RSH concentrations will improve empirical models in phytoplankton uptake studies and will form a new basis for modeling MeHg incorporation in aquatic food webs under various environmental conditions.”

Try to work some of the major numbers from the paper into the abstract. The finding on line 141 that the DOM-RSH (nM) thiols varied by a factor of 40 while the DOC only varied by a factor of 5 was a key one for me, but is not really reflected in the abstract.

The paper also emphasized that Cl- speciation is not controlling MeHg bioavailability, but this finding is not emphasized in the abstract. Instead, the abstract says that salinity ranged from 0.5 to 36 ppt (check the units here). Stating the salinity in the abstract makes it sound like salinity is important, when from my reading of the paper, it is really about the DOM composition.

Response:

We agree with these remarks and have revised the text accordingly, including removing statements about salinity in the abstract:

Line 19:

“Across 20 study sites, DOM-RSH concentrations decreased 40-fold from terrestrial to marine environments whereas dissolved organic carbon (DOC), the typical proxy for MeHg binding sites in DOM, only had a 5-fold decrease.”

Work on summarizing the results, not just what was studied. For example, line 18 states, “We show that the concentration of DOM-associated thiol functional groups (DOM-RSH) dictates MeHg speciation and bioavailability across the terrestrial to marine aquatic continuum.” It would

be better if you gave the direction of the relationship (and maybe even the magnitude), as in, “As the RSH-DOM ratio decreases from terrestrial to marine systems, there is a resulting decrease in MeHg bioavailability to phytoplankton.”

To help illustrate how the abstract could be improved, see the classic piece by Kenneth Landes, A Scrutiny of the Abstract. I’m including the link here in case it is of use to graduate students/post docs involved with this project:

https://www.caam.rice.edu/~symes/CAAM600/abstract_scrutiny.pdf

Response:

We agree and have revised the text.

Line 21:

“MeHg accumulation into phytoplankton was shown to be directly linked to the concentration of specific MeHg binding sites (DOM-RSH), rather than DOC. Therefore, MeHg bioavailability increases systematically across the terrestrial-marine aquatic continuum as the DOM-RSH concentration decreases.”

Introduction

The introduction is very clearly framed and succinctly summarizes the existing literature.

Response:

Thank you for the comment. We have, nevertheless, revised to the second part of the Introduction (lines 60-83), mainly to further clarify the aims and to state the main findings of our study.

Line 27. Work on succinctness. For example, “It is estimated that anthropogenic Hg emissions have caused a three-fold increase in Hg content in surface marine waters compared to pre-industrial conditions . . .” could become, “Anthropogenic Hg emission have caused a three-fold increase . . .”

Response:

We agree on the usefulness of keeping the text succinct. In this particular example, we do want to acknowledge that the three-fold increase is associated with a substantial uncertainty. We therefore prefer to keep “It is estimated that” in this specific case.

Results

Line 86. Remove the comma after et al., in “. . . Dittmar et al., (2008) and . . .”

Response:

Removed as suggested.

Break up the long paragraph that begins on line 80. Lines 80 – 89 are a brief summary of methods. Start the results of the PCA analysis in a new paragraph and improve the topic sentence on line 90 so that it is clear that you will have a paragraph just on the spectral data. Use a different paragraph for the S speciation data.

Response:

We agree with the reviewer’s remark and have split the paragraph into three separate ones and revised the text accordingly. In response to a remark from Reviewer 2, we have added information about the DOM extraction method in the first paragraph (lines 94-100).

The second paragraph now deals with the fluorescence spectral results only and the text at the beginning of the paragraph has been revised to:

Line 107:

“One PCA focused upon the fluorescence spectral data (Figure 2a, b). The first principal component of this PCA described 61% of the data variability and separated samples containing more recently produced fluorescent material (C360) from those with relatively older material (C450).”

The third paragraph deals with the sulfur speciation results and the text at the beginning of the paragraph has been revised to:

Line 117:

“Another PCA was made to visualize the chemical speciation of dissolved organic sulfur by location given its role in MeHg binding (Figure 2c, d). The first two principal components described 79% of the data variability (with 52% in the first principal component) and the sites were more strongly separated in this PCA compared to the one based on fluorescence spectral data.”

Lines 90 – 96. Were there two separate PCA analyses done here? One on the spectral data and one on the S speciation? Some of the language in the text is confusing. For example, line 95 states, “The second PC described 27% of the data variability and discriminated fluorescent DOM by origin (C390 bacterially derived material, C520 terrestrially dominated material).” I think this is the first PCA analysis, but the second PC, but I had to closely study Fig. 2 to check this. If indeed there are two separate PCA analyses done here, describing each in its own paragraph would help considerably. You may even want to consider subheaders, such as paragraph header.

Response:

We agree that this could appear confusing and we have spitted the text into two separate paragraphs as described in our repose to the preceding comment. We have also omitted the abbreviation “PC” and instead spell out “principle component” for further improved clarity.

OK, as I’m reading along to line 103, I am seeing that there are two PCAs here. Not introducing the second PCA on line 92 would help. Wait and introduce the second PCA and Figure 2c and d on line 103 when you give the results for this PCA.

Response:

We agree and have revised the text accordingly as described in our responses to the two preceding comments.

Line 112 – the huge variability between the estuarine sites is really interesting here and the explanation about mixing between both terrestrial and marine algal DOM makes sense

Response:

Thank you. Yes, we agree this variability is interesting and quite well-captured both by the sulfur speciation and fluorescence spectral results.

Lines 146 – 168 – break up this long paragraph, perhaps by starting a new summary paragraph on line 161, with “Combined, these results. . .”

Response:

We have substantially revised this paragraph to decrease its length and clarify its overall point and we have split the paragraph as suggested to reduce the length of each paragraph.

Line 167. Correct punctuation is “. . . experiments are, however, not straightforward. . .”

Response:

Revised as suggested.

Line 201. The two isotopes were a nice approach to determine that the non-exchangeable pool of MeHg was very small.

Response:

Thank you. Yes, this is a powerful approach to investigate potential non-exchangeable pools.

Line 244. Can you put the reference to the relevant figure or table at the end of this sentence?

Response:

Thank you, we have added a reference to Figure 4 and the end of the sentence (line 262).

Line 263. Ah, it is so nice to see all of the variability in MeHg VCFs explained in terms of DOM-RSH.

Response:

Thank you, we are excited about these results.

Line 270. Oh no, it is hard to even begin to read this giant paragraph. Use paragraphs of 3 – 7 sentences to help your reader! Consider paragraph breaks on line 277 with the thermodynamic modeling. Make line 285 saying that MeHg speciation is not driven by Cl⁻ concentration the topic sentence of a paragraph on your thermodynamic results. The importance of Cl⁻ has been brought up enough in the literature that this result should be easy to find in the topic sentence. Can you move some of this material into the discussion? Some of the interpretation, such as that the role of Cl⁻ in controlling MeHg speciation has been overstated, sounds like it is more discussion than results.

Response:

We agree this paragraph was too long and we have split it into four separate ones (lines 309-356). Some of the text has been revised to acknowledge that uptake of MeHgCl may still be significant even if this species constitutes a minor fraction of the total dissolved MeHg concentration (lines 3325-336).

We propose to use the following as topical sentences for MeHg speciation not being driven by Cl⁻ concentration:

Line 333:

“The small variability in MeHg speciation (including the magnitude of the MeHgCl fraction) was thus primarily driven by differences in the DOM-RSH concentration among sites and not by Cl⁻. This result highlights DOM-RSH concentration as the major control of MeHg availability for cellular uptake, regardless of the exact uptake mechanism.”

We further agree on the remark regarding Result text and Discussion text. After evaluating all comments from the reviewers, we believe that the best way to present this study is to use a combined Results and Discussion section. We note that either separated or combined

Results and Discussion sections are used in Nature Communication publications. We therefore propose a combined Results and Discussion section and to reformat the previous “Discussion” section into “Implications” as a sub-section of the R&D. Following this rearrangement, we have removed some repetitive text and instead expanded the discussion around the conceptual Figure 5 in the new Implications paragraph. These revisions are described in detail in response to specific remarks below.

Line 297. I can't wait to see your next study, where you include the open ocean. But again, I think this material is discussion, not results.

Response:

We agree and after evaluating all comments from the reviewers we believe that the best way to present this study is to use a combined Results and Discussion section, as described in our response to the preceding remark.

Line 309. Remember to offset “however” with commas.

Response:

We have partly revised this text and removed the word “however”.

The extensive use of acronyms in this manuscript don't help with readability. For example, N-Acetyl-Penicillamine is abbreviated Nacpen. Even though this acronym is used a lot during the manuscript, it was not one that I knew so it complicated the reading. Completing ligand exchange is CLE. While some of these acronyms (e.g., DOM, BCF, etc.) are quite common in the field, I had to repeatedly refer back to others. Could you systematically go through this manuscript and try to remove a couple of the less common and/or less frequently used acronyms so that the paper reads less like alphabet soup?

Readability is also hampered by some of the notation. For example, the paper refers both to the RSH/DOC ratio and also to the DOM-RSH, which is not a ratio, but is the concentration of DOM-associated thiol groups (line 19). Is there some clearer way to display DOM-RSH so the poor reader is not getting mixed up between the ratios and the concentration of thiol groups? Maybe just call it thiol concentration?

Response:

We agree that it's important to avoid excessive use of abbreviations and unclear notations. We have removed the use of the following abbreviations and instead spell out the full text:

PC: principle component

DOS: dissolved organic sulfur

IHSS: International Humic Substances Society

OrgS_{red}: reduced organic sulfur

CLE: competing ligand exchange

The notation “DOM-RSH” is central for the content of the study, not only in the text but also in chemical reactions and mathematical equations (which is also the case for “Nacpen”). We therefore believe that it is best to consistently keep the notation “DOM-RSH” and “Nacpen” throughout the manuscript. We hope that the descriptions and definitions of the terms “RSH/DOC ratio” and “DOM-RSH concentration” on lines 136-139 improves readability.

Discussion

Line 319. It should be “. . . how DOM controls MeHg bioavailability. . .”

Response:

Revised as suggested.

Line 319. I think you are missing the word “of” and the tense is wrong. It should be “. . . a quantitative assessment of how DOM controls MeHg bioavailability. . .”

Response:

Revised as suggested.

Lines 344 – 368. Can you separate out your discussion of how MeHg is taken up by phytoplankton cells and implications for Fig. 4 from your discussion of Fig. 5.

Response:

Yes, we agree that it is useful to separate the discussions of Fig 4 and Fig 5. We have now rearranged the text so that Fig 4 is discussed in the section “Bioavailability and VCF of MeHg depend on the total concentration of DOM associated thiols” and Fig 5 is discussed in the section “Implications”.

Figure 5 is a very interesting figure and nicely summarizes the paper, but is only mentioned briefly on lines 345 and 351. For such an important visual summary, this figure merits more play. Discuss it in its own paragraph. Also discuss it in light of the literature. For example, do field studies support the idea that shelf systems have the highest MeHg in algal cells? Does this translate up the food chain? And how does it tie in with the common observation that oligotrophic lakes often have very high MeHg concentrations in fish?

Response:

We agree that Figure 5 should be better highlighted in the manuscript and have addressed this by three revisions. First, we have split the figure into three separate panels to more clearly communicate the three key messages from our study on how DOM-RSH controls MeHg bioavailability. These are now numbered 1., 2. and 3. in the figure. Second, by merging the Results and Discussion sections we have moved discussions not concerning Figure 5 from the new “Implications” paragraph to other sections of the manuscript. The new “Implications” section thus fully focuses on Figure 5. Third, we have added discussions on how field studies support increased MeHg concentration in seston from systems with higher levels of marine DOM over terrestrial DOM.

Line 366:

“Increased MeHg concentration in seston from systems with higher levels of marine DOM over terrestrial DOM has been demonstrated through field observations in terrestrial 19, estuarine, and marine aquatic systems 17,54,55 as well as in laboratory uptake experiments 18. Where these previous studies invoke various principles in effort to explain differences in uptake, such as higher aromaticity 19, larger molecular weight 17, and speciation with inorganic ligands 14,18 we can now more clearly link MeHg uptake to the DOM-RSH concentration. Specifically, our results, including the uptake experiment, suggest that changes in bioavailability are driven by MeHg speciation with DOM-RSH, which is not well predicted by local DOC concentrations.”

Line 374. Spell out what you mean by “potentially opposite impacts on MeHg formation and bioaccumulation.” Which anthropogenic alteration is likely to have which effect?

Response:

We agree with the remark and we have removed the statement “potentially opposite impacts...” and instead directly describe the expected effects of specific processes/scenarios.

Line 376:

“Environmental change scenarios are expected to directly impact DOM loading to aquatic ecosystems, and therefore MeHg bioavailability⁵⁶. For instance, eutrophication leads to the increased production of autochthonous DOM ^{57,58} which tends to produce DOM with low RSH/DOC. Allochthonous, terrestrial DOM inputs to aquatic systems are impacted by changes in watershed runoff due to e.g. climate change or land use and tend to deliver DOM with high RSH/DOC to coastal systems. Based on our results, increased proportions of fresh, marine DOM would lead to disproportionately smaller increases in local DOM-RSH than if terrestrially derived DOM increased in a system. The use of DOC as a proxy for MeHg binding capacity would thus underestimate MeHg availability in algal blooms and overestimate it for increased terrestrial runoff.”

Line 378. Are you anticipating an increase in DOC overall or a change in the RSH/DOC ratio? Would changing the concentration of both RSH and DOC without changing the ratio change VCFs?

Response:

We realize that the text regarding DOC, DOM-RSH and RSH/DOC ratio was not very clear in this section and we have largely rewritten it.

Line 376:

“Environmental change scenarios are expected to directly impact DOM loading to aquatic ecosystems, and therefore MeHg bioavailability⁵⁶. For instance, eutrophication leads to the increased production of autochthonous DOM ^{57,58} which tends to produce DOM with low RSH/DOC. Allochthonous, terrestrial DOM inputs to aquatic systems are impacted by changes in watershed runoff due to e.g. climate change or land use and tend to deliver DOM with high RSH/DOC to coastal systems. Based on our results, increased proportions of fresh, marine DOM would lead to disproportionately smaller increases in local DOM-RSH than if terrestrially derived DOM increased in a system. The use of DOC as a proxy for MeHg binding capacity would thus underestimate MeHg availability in algal blooms and overestimate it for increased terrestrial runoff.”

Line 385:

“Our study highlights that any environmental process causing an increase in specifically the molar concentration of DOM-RSH will decrease MeHg bioavailability and bioaccumulation. This effect is expected to be large at low to moderate nanomolar-level concentrations of DOM-RSH ($\leq \sim 25$ nM in our uptake experiment, Figure 4) and minor at higher concentrations.”

Line 380. How would an algal bloom change the RSH/DOC ratio? Why would terrestrial runoff increase RSH/DOC ratio?

Response:

We hope that these remarks are addressed by our response and revisions to the two preceding remarks, i.e. the revisions on lines 376 and 385. Our point is that any process causing an increase in the molar concentration of DOM-RSH (algal bloom or terrestrial runoff) will decrease MeHg bioavailability and bioaccumulation. However, the use of DOC as a proxy would underestimate MeHg availability in algal blooms and overestimate it for increased terrestrial runoff since the RSH/DOC ratio would decrease following algal blooms but increase following increased terrestrial runoff.

The results of the PCA analysis are never mentioned in the discussion. How does the PCA analysis relate to the material in the discussion on MeHg bioavailability to phytoplankton?

Response:

We address this remark by combining Results and Discussion into a joint section as described in response to previous remarks and by the revisions made in response to the above remarks on the PCA analysis paragraphs. The purpose of the PCA analysis is to provide an overview descriptive presentation of key features of the DOM from the different sites and support the classification of sites into four groups. We hope that this is clear in the manuscript following the revisions made.

Overall, spend some more time on the implications of the importance of the DOM-associated thiols. I think what I'm hearing is that measuring DOC is not really sufficient to predict bioavailability to phytoplankton. Instead, XANES is needed to quantify the number of thiol groups associated with the DOM. But most researchers don't have XANES! So what would you recommend for researchers seeking to study global environmental change and for monitoring programs that still use DOC?

Response:

It should be noted that we quantified thiols by LC-MS/MS (recently developed method by Liem-Nguyen et al. 2019) and not by XANES. We used XANES for the extended characterization of sulfur speciation of the DOM samples (presented in Figure 2). We also included XANES results (the total organic reduced sulfur content) in our comparison with previous studies (lines 162-179) which have used synchrotron radiation-based methods and expressed the total thiol content per amount of total organic reduced sulfur. But XANES is not required to quantify total thiol content expressed as DOM-RSH molar concentration or RSH/DOC ratio as done in this study. LC-MS/MS is nowadays a standard equipment in analytical chemistry labs and thus much more accessible than synchrotron-based methods, which should facilitate to include total thiol measurements in future studies. We have added the following text to highlight this aspect.

Line 177:

“Furthermore, the recently developed LC-MS/MS method used here to determine total thiol concentrations 35 is more accessible, and requires less sample amounts, than EXAFS and XANES techniques and will facilitate an expansion of such measurements in future studies.”

Methods

Line 418 sounds like it is statistical methods – separate it out into a new paragraph.

Response:

The data processing approaches described from Line 431 are specific for the fluorescence data only and we therefore think it is most suitable to keep all this information as one paragraph.

Check punctuation on line 439.

Response:

We believe the punctuation is now correct.

Lines 479- 515. Break up this long paragraph.

Response:

We have split the paragraph into three paragraphs.

Line 499. What was the general magnitude of the DOM blank that had to be subtracted

Response:

The mean and median DOM blank subtracted were 15.9 nM and 9.0 nM, respectively, and 10 samples were blank subtracted. This is now indicated in the text, line 516:

“Therefore, the DOM blank was subtracted from the MeHg(Nacpen) values when elevated in respect to the GC-ICPMS measurement, which included ten DOM experiments (blank range = 1.2 – 62 nM, mean = 15.9 nM).”

Line 564. Salinity units of ppt are not appropriate -see comments under Table 1.

Response:

We agree and have switched the salinity unit to psu throughout the manuscript.

Table 1.

Salinity in ppt is a rather old notation and not consistent with the use of a salinometer to measure it. When measured with a meter, salinity is measured as conductivity relative to a standard, which makes it unitless. Those who hate unitless numbers use practical salinity units (psu). In contrast, latitude and longitude do have units, which are not currently shown in the table.

Response:

We agree and have switched the salinity unit to psu throughout the manuscript.

Figure 1

Please add a few place names to the map, to orient people who are not very familiar with the East Coast of the United States. Also consider the legibility of this map when printed in black and white or viewed by people who are color blind. In addition to using colors for habitat type, use different symbols (e.g., triangles, squares, etc.) to distinguish between these habitats.

Response:

We agree and have revised the figure according to the reviewer comments, more labels were added and the site markers vary in shape by system type.

Figure 2

Please make the legend and axes and site names larger. I can hardly read them, even blown up 150% and when they are published in the journal, they are going to be even smaller.

Label some of the individual panels, such as with Fluorescence spectral data and the other with S speciation. The current labeling just says, “variables – PCA” which is not particularly helpful.

Improve this heading. Write out the x-axis labels in panels b and d. I'm assuming that Dim1 is dimension 1? You are really making the reader work hard to interpret these panels. Write out axes labels and use the text and caption to help the reader interpret the figure. The addition of the oxidation states on panel d definitely helps, but add a sentence about the dimensions, particularly linking the dimensions, the loadings, and the contributions. I do not understand the scale of the "contributions" on panels b and d. I am used to thinking about component loadings ≥ 0.6 or ≤ -0.06 as being important but not really sure where the info for "contrib" came from, and I'm not getting that from the text, either.

Response:

The text associated with Figure 2 a-d have all been increased to make the figure easier to read. We have decided to not label the panels, but instead include that description in the figure legend. The loading plots (b and d) were redone to remove the z-axis coloration, which used to represent the variable contribution to the PC. We felt that this information was not necessary and only made the figure more confusing. Regarding the dimensions, the R function we used defaults to the axis labels "dim1" and "dim2", but it is just PC1 and PC2. The axis labels were changed for clarity.

Figure 3

Looks great. Very interesting progression in DOM thiol concentrations in panel 3a.

Response:

Thank you, yes we think this trend indeed is very interesting.

Figure 4

Put the name of the diatom species in the caption. Does this relationship still hold even if the point in the top-right corner of panel a is removed? Which points are which? It would be nice to see which points in figure 4a correspond to the points in 4b. Since there are so few points, coding them by using different symbols would be reasonable.

Response:

As we mentioned in the summary description of our revision work we are very grateful for the remark about the data point in the top-right corner of the previous Figure 4 as this turned out to be a calculation error. The revised VCF for this data point (lowest DOM-RSH and DOC concentrations for the River site) is 8.3×10^4 compared to the previous value of 21×10^4 . As can be seen in the revised Figure 4, this rectification does not alter the fact that DOM-RSH concentration is a much stronger predictor of MeHg VCF than DOC concentration is. Further, the relationship between VCF and DOM-RSH is still best fitted by a power function, but with a power coefficient of -0.4 compared to -1 in the previous model. Therefore, we believe it is most appropriate and informative to plot VCF versus concentrations of DOM-RSH and DOC instead of versus $1/\text{DOM-RSH}$ and $1/\text{DOC}$ as was done in the original figure. In this way, Figure 4 also has the same basic layout as figures of VCF versus DOC in previous studies and will thus be more familiar to many readers. Furthermore, we have coded the data points in Figure 4 with different symbols as suggested by the reviewer.

Although the strong empirical relationship between VCF and DOM-RSH concentration holds also for the revised data set we have revised and expanded the mechanistic modeling of the relationship. As we discuss in the revised manuscript, based on current theory, the observed relationship can in principle be explained by two cases: (A) a reversible equilibrium uptake model (the model we presented in the original manuscript), or (B) a

quasi-irreversible kinetic uptake model. We discuss both these cases and present them in further detail in the supporting information (Figure S5). Case A fits the experimental data fairly well but does not seem to fully capture the complex process of MeHg uptake. Case B fits the data well but we nevertheless conclude that further mechanistic studies on cellular internalization of MeHg are required to better parameterize uptake as a direct function of MeHg speciation, cellular abundance, and cell size.

In the section “MeHg speciation and bioavailability in natural systems”, we discuss that regardless of the exact uptake mechanism (Case A or Case B), the availability of MeHg for uptake is controlled by the concentration of DOM-RSH.

Line 280:

“Based on current theory, two cases can be invoked to explain the power function relationship between MeHg VCFs and DOM-RSH concentrations (Figure S5). The DOM-RSH concentration may control the equilibrium partitioning between MeHg bound to DOM-RSH and accumulated in plankton cells via reversible uptake mechanisms (case A). Alternatively, it may control the equilibrium distribution between MeHg(DOM-RS) and inorganic MeHg complexes (e.g. MeHgCl) in solution which are accumulated in cells with species-specific rates via quasi-irreversible mechanisms (case B). Case A presents MeHg cellular uptake as a thermodynamic equilibrium function between one MeHg species in solution (MeHg(DOM-RS)) and one associated with cells (MeHg(cell)_{n+1}) with a conditional constant K₈ (Eqn S1d). Rearranging the equilibrium expression of MeHg between cellular ligands and DOM-RS results in the MeHg VCF (i.e. [MeHg(cell)_{n+1}] / [MeHg(DOM-RS)]) being proportional to 1/[DOM-RSH] assuming the term K₈[H⁺][cell] is constant under the experimental conditions (full equations in SI). With the limited data generated for this study, the empirical relationship between VCF and 1/[DOM-RSH] is not linear in the space as was expected (Figure S5), thus the equilibrium model likely does not fully capture the complex process of MeHg uptake.”

Line 294:

“Case B accounts for a simplified, simultaneous uptake of the dominant organic and inorganic forms of MeHg in the experimental assays (MeHg(DOM-RS) and MeHgCl, respectively) based on their unique uptake rate constants and specific concentrations. Prior studies denote a faster uptake of MeHgCl due to passive diffusion across the cell membrane 22 while uptake of MeHg(DOM-RS) complexes are thought to be slowed by interactions at the cell surface prior to internalization 23. The uptake rate constant of MeHg(DOM-RS) was estimated based on the highest DOM exposure concentration used in this study at 0.004 amol μm⁻³ hr⁻¹ nM⁻¹. The MeHgCl uptake rate constant was set to fit the data, resulting in a value of 0.5 amol μm⁻³ hr⁻¹ nM⁻¹; at the high end of published values (e.g 0.175 12 and 0.5 51). The Case B modeling results are plotted in Figure S5 against the RSH concentration of the exposure medium. The model fit the results well, but further mechanistic studies on cellular internalization of MeHg are required to better parameterize uptake as a direct function of MeHg speciation, cellular abundance, and cell size. Based on our modeling we do, however, suggest that bioaccumulation of MeHg into phytoplankton needs to be determined using a combined thermodynamic-kinetic model that considers water column MeHg speciation and the differences in the uptake rate constants for the different MeHg complexes.”

Line 333:

“The small variability in MeHg speciation (including the magnitude of the MeHgCl fraction) was thus primarily driven by differences in the DOM-RSH concentration among sites and not by Cl-. This result highlights DOM-RSH concentration as the major control of MeHg availability for cellular uptake, regardless of the exact uptake mechanism.”

Figure 5

Would it make sense to show the RSH/DOC ratio to illustrate how it changes across the continuum of sites? The little squares that link from the bottom right or the bottom left or the middle of the bar charts to the cartoons are weird and not adding anything.

Response:

We have considered several versions of figure 5 to clearly communicate the core messages of the study. We propose a revised figure where the three key points 1., 2. and 3. regarding the role of DOM-RSH in controlling MeHg bioavailability are visualized individually in separate panels. We have split the left-hand panel of the original figure 5 into two separate ones. We have further revised the orientation of the cartoon map in panel 2. to more clearly visualize the aquatic gradient. We have also revised the layout of panel 3. to match the revised figure 4. Notably, the distribution of MeHg between DOM-RSH and plankton cells in panel 1. may progress via a direct exchange between these two phases (the Case A model) or via a distribution of MeHg between MeHg(DOM-RS) and MeHgCl complexes (the Case B model).

We have considered to add also the RSH/DOC ratio into figure 5 but we think the figure then becomes rather messy and difficult to read.

Reviewer #2 (Remarks to the Author):

The manuscript is entitled "Dissolved organic matter thiol concentrations determine methylmercury bioavailability across the terrestrial-marine aquatic continuum"

General comments:

This is a manuscript which deals with the role of the DOM-Thiol groups for the complexation and bioavailability of MeHg across different aquatic systems. The authors determined the thermodynamic and kinetic interactions between MeHg and DOM-Thiol groups extracted from varied aquatic sites. The results are quite interesting to me to see such a good relationship between RSH concentrations and VCF MeHg in phytoplankton (Anyway, is there any relationship between them if we removed only one data of shelf sea in Fig. 4a). Additionally, the observation that K doesn't change too much is pretty valuable. However, one big gap is that the DOM they are studying is high molecular weight...so, this is only a part of the story. The strategy of the sampling also lacks the shelf data although the trends with that point make sense. Nature Comm is short like Nature. This seems like a pretty long paper, full of small details. I think it's well done and pretty important. I indeed learned a lot from this paper related to potential mechanisms How MeHg gets into the cell of phytoplankton.

Response:

We thank the reviewer for the overall positive feedback, and the valuable specific remarks. We address the remark regarding high molecular weight DOM in our response to the reviewer's specific remark on this topic below. We agree that more sampling sites at the shelf would have been ideal but that the trends in the results are coherent. Regarding the length of the paper and technical details, we have removed some technical details and added text to better describe the broader implications of the study. These revisions are described in detail in response to remarks from reviewer 1.

As we mentioned in the summary description of our revision work we are very grateful for the remark about the data point in the top-right corner of the previous Figure 4 as this turned out to be a calculation error. The revised VCF for this data point (lowest DOM-RSH and DOC concentrations for the River site) is 8.3×10^4 compared to the previous value of 21×10^4 . As can be seen in the revised Figure 4, this rectification does not alter the fact that DOM-RSH concentration is a much stronger predictor of MeHg VCF than DOC concentration is. Further, the relationship between VCF and DOM-RSH is still best fitted by a power function, but with a power coefficient of -0.4 compared to -1 in the previous model. Therefore, we believe it is most appropriate and informative to plot VCF versus concentrations of DOM-RSH and DOC instead of versus $1/\text{DOM-RSH}$ and $1/\text{DOC}$ as was done in the original figure. In this way, Figure 4 also has the same basic layout as figures of VCF versus DOC in previous studies and will thus be more familiar to many readers. Although the strong empirical relationship between VCF and DOM-RSH concentration holds also for the revised data set we have revised and expanded the mechanistic modeling of the relationship. As we discuss in the revised manuscript, based on current theory, the observed relationship can in principle be explained by two cases: (A) a reversible equilibrium uptake model (the model we presented in the original manuscript), or (B) a quasi-irreversible kinetic uptake model. We discuss both these cases and present them in further detail in the supporting information (Figure S5). Case A fits the experimental data fairly well but does not seem to fully capture the complex process of MeHg uptake. Case B fits the data well but we nevertheless conclude that further mechanistic studies on cellular

internalization of MeHg are required to better parameterize uptake as a direct function of MeHg speciation, cellular abundance, and cell size.

In the section “MeHg speciation and bioavailability in natural systems”, we discuss that regardless of the exact uptake mechanism (Case A or Case B), the availability of MeHg for uptake is controlled by the concentration of DOM-RSH.

Line 280:

“Based on current theory, two cases can be invoked to explain the power function relationship between MeHg VCFs and DOM-RSH concentrations (Figure S5). The DOM-RSH concentration may control the equilibrium partitioning between MeHg bound to DOM-RSH and accumulated in plankton cells via reversible uptake mechanisms (case A). Alternatively, it may control the equilibrium distribution between MeHg(DOM-RS) and inorganic MeHg complexes (e.g. MeHgCl) in solution which are accumulated in cells with species-specific rates via quasi-irreversible mechanisms (case B). Case A presents MeHg cellular uptake as a thermodynamic equilibrium function between one MeHg species in solution (MeHg(DOM-RS)) and one associated with cells (MeHg(cell)_{n+1}) with a conditional constant K₈ (Eqn S1d). Rearranging the equilibrium expression of MeHg between cellular ligands and DOM-RS results in the MeHg VCF (i.e. [MeHg(cell)_{n+1}] / [MeHg(DOM-RS)]) being proportional to 1/[DOM-RSH] assuming the term K₈[[H⁺][cell]] is constant under the experimental conditions (full equations in SI). With the limited data generated for this study, the empirical relationship between VCF and 1/[DOM-RSH] is not linear in the space as was expected (Figure S5), thus the equilibrium model likely does not fully capture the complex process of MeHg uptake.”

Line 294:

“Case B accounts for a simplified, simultaneous uptake of the dominant organic and inorganic forms of MeHg in the experimental assays (MeHg(DOM-RS) and MeHgCl, respectively) based on their unique uptake rate constants and specific concentrations. Prior studies denote a faster uptake of MeHgCl due to passive diffusion across the cell membrane 22 while uptake of MeHg(DOM-RS) complexes are thought to be slowed by interactions at the cell surface prior to internalization 23. The uptake rate constant of MeHg(DOM-RS) was estimated based on the highest DOM exposure concentration used in this study at 0.004 amol μm⁻³ hr⁻¹ nM⁻¹. The MeHgCl uptake rate constant was set to fit the data, resulting in a value of 0.5 amol μm⁻³ hr⁻¹ nM⁻¹; at the high end of published values (e.g 0.175 12 and 0.5 51). The Case B modeling results are plotted in Figure S5 against the RSH concentration of the exposure medium. The model fit the results well, but further mechanistic studies on cellular internalization of MeHg are required to better parameterize uptake as a direct function of MeHg speciation, cellular abundance, and cell size. Based on our modeling we do, however, suggest that bioaccumulation of MeHg into phytoplankton needs to be determined using a combined thermodynamic-kinetic model that considers water column MeHg speciation and the differences in the uptake rate constants for the different MeHg complexes.”

Line 333:

“The small variability in MeHg speciation (including the magnitude of the MeHgCl fraction) was thus primarily driven by differences in the DOM-RSH concentration among sites and not by Cl⁻. This result highlights DOM-RSH concentration as the major control of MeHg availability for cellular uptake, regardless of the exact uptake mechanism.”

Some minor comments:

=====

Line 118, “the chemical speciation and bioavailability of dissolved MeHg...” This is kind of a hypothesis. To be sure, this paper helps support this hypothesis, but this statement sounds too categorical.

Response:

We agree that this statement was too categorical and have revised it.

Line 133:

“The binding capacity and binding affinity of in situ MeHg ligands are important thermodynamic parameters influencing the chemical speciation and bioavailability of dissolved MeHg in natural waters.”

Line 122 “DOM binding capacity” is a little too ambiguous a term in this context. Since it is DOC normalized, consider using something like “DOM-specific binding capacity” or something.

Response:

We agree that the term “DOM binding capacity” may appear slightly ambiguous. After all, also DOM functional groups other than thiols can bind MeHg, for example amines or carboxyl groups, even though these are hardly relevant for natural conditions. Since we clearly define (lines 136-139) our use of the term “binding capacity”, and since we define and use it both for DOM and *in situ* in water at a given site, we believe that adding a term like “-specific” or similar may not resolve potential ambiguousness. Instead we propose to specify that our definition of binding capacity is contextual for the present study by revising the text to, line 134:

“Here, we define the MeHg binding capacity as the...”

Line 129-132, “binding affinity...” citation for this method?

Response:

Thank you, we had missed to insert the proper citation here, we only included it in the Methods section. We have now added to line 145 a citation to Liem-Nguyen et al. 2017 from which the competing ligand exchange methodology was adapted.

Line 133, how do these results compare to total Hg? Wouldn't we expect that their concentrations would be similar?

Response:

This is a very interesting question. We do expect that the total thiol concentration would be the same but that MeHg forms one-coordinated and Hg(II) forms two-coordinated complexes with thiols. One could hypothesize that not all thiol groups are available for two-coordinated binding to Hg(II). However, the few studies so far which have investigated this have shown that, at least for the Suwanee River DOM, the binding of Hg(II) can be explained with a model that assumes 100% of the thiol groups are available for formation of two-coordinated complexes with Hg(II) (Song et al. ES&T 2018). In contrast, for thiols associated by bacterial membranes (of *Geobacter Sulfurreducens*) only approximately 5% of the total concentration of thiol groups are located closely enough to form two-coordinated Hg(RS)₂ complexes, while the remaining 95% can only form ternary complexes with one membrane-thiol ligand and with either one other membrane ligand (containing O or N functional groups) or a dissolved thiol compound (Song et al. ES&T 2020). This is indeed an interesting area for further research.

Line 199, the ambient ligands come out looking like Nacpen...this is what happens using CLE. Any comments on analytical “windows” that are created by this approach?

Response:

We are not sure we fully understand the remark, and we apologize if we have misunderstood, but for our study there are two important aspects of analytical windows with the CLE approach. These aspects are the range in log K captured by the approach (for the thermodynamic experiments) and the measurement frequency at which data points can be generated (for the kinetic experiment).

Regarding the range in log K that can be quantified for DOM samples, this is determined by the log K of the competing ligand (Nacpen in our case) and there is a limitation in how large differences from this value that can be determined. The approach we use was adapted from Liem-Nguyen et al. Environ. Chem. 2017 who originally developed the method for inorganic Hg(II). As demonstrated in that study, the applicable range in log K was rather large and Hg(II) complexes both with thiols and with iodide could be determined despite several orders of magnitude differences in stability constant. It is also important to note that if the log K for a specific sample falls “outside” of the analytical window and cannot be accurately quantified due to a largely deviating log K, this will be detected with the use of this methodology. Such a situation will be manifested by all MeHg being bound either to the sample ligands or to the competing ligand (if the log K of the sample ligands are considerably higher, or considerably lower than that of the competing ligand) across a wide range in molar ratio of the two ligand pools. Notably, this was not the case for any of the DOM samples in this study.

Regarding the “kinetic window” this is constrained by the transfer handling of samples from the glovebox (where incubations are done) to the analytical instrument and, mainly, by the analysis time of the LC-ICPMS method. With the LC-ICPMS method we used the maximum measurement frequency is limited to approximately 5 min. As discussed in the manuscript (lines 230-236), this was sufficient to conclude that kinetic constraints was not a major process controlling differences in MeHg bioavailability among the DOM samples. However, the comparably low measurement frequency did not enable to resolve more subtle potential differences between the DOM types, which remains an open and interesting research question.

Line 224, complexation for MeHg sounds faster than for Hg(II)...any comments?

Response:

Yes, studies on MeHg and Hg(II) ligand exchange kinetics, although being relatively few, consistently point to faster kinetics for MeHg. The understanding of these processes, however, seems not yet sufficiently detailed to fully explain the differences quantitatively. Presumably the differences are partly caused by Hg(II) forming intermediate complexes with one exchanged ligand, and also plausibly because Hg(II) binding is more likely to induce structural rearrangements of the ligand and chelating effects.

Line 261, “The results demonstrate that the total DOM-RSH...” The word “demonstrate” is a little too strong here. Change to “the results are consistent with the hypothesis that...”

Response:

We agree and have revised the text accordingly.

Line 271:

“The results support that the total DOM-RSH concentration is a primary controlling factor of MeHg bioavailability and cellular accumulation by phytoplankton.”

Line 394, talk about the limitations of Dittmar approach...this is mostly high molecular weight material. By ignoring LMW thiols, the importance of complexation is underestimated a bit. But, LMW thiols have been reported to be especially bioavailable...so, the MW/LW distinction might be important to discuss.

Response:

We agree that the solid-phase extraction of DOM is an important aspect, and we thank the reviewer for the remark. Regarding HMW and LMW thiols and DOM, we note that ultrafiltration is the technique which is commonly used to selectively isolate the HMW fraction of DOM (typically > 1kDa) and it is estimated that ten to twenty percent of DOM falls into this high-molecular mass fraction (Catala et al., Appl. Microbiol. Biotechnol. 2021). In comparison, the PPL-SPE used in our study (developed by Dittmar and being the most commonly used SPE approach to extract DOM) extracts a broad range of DOM compounds with respect to size and hydrophobicity and thus a more representative fraction of the DOM than ultrafiltration does (Dittmar et al., Limnol. Ocean. Methods 2008, Li et al., Anal. Chem. 2016). However, the SPE recovery does vary with chemical properties of the DOM compounds and, as pointed out by the reviewer, the extraction efficiency with PPL-SPE is lower for highly hydrophilic low-molecular-weight compounds (Dittmar et al., Limnol. Ocean. Methods 2008; Ksionzek et al., Ocean Sci. 2019; Lechtenfeld et al., Depp Sea Res. I 2011; Pohlabeln et al., Front. Mar. Sci. 2017; Raeke et al., Environ. Sci. Proc. Imp. 2016). In the context of our study, we believe that these limitations associated with the PPL-SPE approach to isolate DOM do not compromise our conclusions, but we fully agree with the reviewer that the potential implications should be discussed in the manuscript. We have added the following text discussing these aspects:

Line 93:

“DOM was extracted from 20-40 L of surface water at each site following Dittmar et al. 26. This solid phase extraction method (based on a styrene divinyl benzene polymer, so called PPL-SPE) is the most common approach for DOM enrichment and desalting, which is a prerequisite for most types of molecular characterizations 27, including the ones in this study. In general, PPL-SPE captures a broad range of compounds and retains overall differences in the molecular composition of the DOM between samples 28–31. However, since external factors such as sample salinity impact the carbon yield of the extraction, we normalized relevant parameters to DOC concentrations to avoid quantitative biases.”

Line 237:

“The results of the kinetic exchange tests could be influenced by the use of extracted DOM since the molecular composition of a water sample can bias the extraction yields of specific compounds. Recent studies suggest that small hydrophilic molecules are partly lost from the sorbent during the extraction process 31. In the context of our work, the potential loss of small hydrophilic DOM-RSH compounds needs to be considered, especially since these compounds form the most bioavailable and easily exchanged MeHg-thiol complexes 12,23. In theory, if small hydrophilic thiols constituted a significant fraction of total DOM-RSH compounds but were lost in the DOM extraction, the in situ turnover for MeHg ligand exchange would be even faster than the <1-3 min we calculated. Therefore, our conclusion that MeHg ligand exchange

kinetics among DOM-RSH compounds is fast and not a major control of MeHg bioavailability would still be valid, or even amplified, if there were losses of small hydrophilic thiols in the SPE isolation.”

Line 463, citation for this method?

Response:

The citation is now given in the first sentence of the paragraphs starting at:

lines 463:

“The MeHg(Nacpen) complex was measured using LC-ICPMS (Perkin Elmer Elan DRC) (adapted from Liem-Nguyen et al. ⁶⁵).”

lines 496:

“Standard competing ligand exchange techniques were used in the overall design of the experiment ⁶⁶, but were refined by the direct quantification of the MeHg complex with the competing ligand, i.e., MeHg(Nacpen), using LC-ICPMS ²⁸ as described above.”

Line 522, citation for the K6 value?

Response:

Thank you, we had missed to insert the citation and have revised the text to:

Line 543:

“Reactions 5 and 6 describe the acid dissociation reaction of the thiol groups in Nacpen and DOM, respectively, and the numerical values of the corresponding constants were taken from Liem-Nguyen et al. ⁶⁵.”

Reviewer #1 (Remarks to the Author):

Overall

This paper has undergone substantial revisions since the last version. The authors have made every effort to address the reviewers' comments and have explained their changes in the author response. Great job with such a thorough and thoughtful response. The revisions have resulted in a manuscript that is much easier to read and has addressed some problems from the last version, such as an incorrectly calculated VCF value. More minor problems, such as missing units, or too many acronyms, have also been addressed. The figures have been improved. For example, the x-axis on Fig. 4 is much more easily interpretable.

The paper has clearly articulated its key finding, mainly that DOC is not a great proxy for the thiol concentration in the DOM, and the latter (DOM-RSH) is key to predicting MeHg bioavailability. This finding helps resolve a long-standing conundrum about why DOM has been found to have variable effects on phytoplankton MeHg accumulation. The limitation of this study is that Fig. 4 uses only 6 data points to show that DOM-RSH is a better predictor for VCF than DOC. But I suppose that future studies will fill in with more data, and this paper will get the ball rolling! Below, I have some comments on specific points in the manuscript. At this point, my comments are mostly minor clarifications, not substantial changes. Overall, nice work!

Abstract

The abstract now does a better job of highlighting key findings and specific magnitudes, such as in the sentence, "Across 20 study sites, DOM-RSH concentrations decreased 40-fold from terrestrial to marine environments whereas dissolved organic carbon (DOC), the typical proxy for MeHg binding sites in DOM, only had a 5-fold decrease." The direction of the effect in relationship to the terrestrial-marine continuum is also clear from the abstract.

Introduction

Line 35. The use of the reference number in superscript right next to 107 is potentially confusing. In addition, the use of the semicolon is not correct. A semicolon is used to connect two complete sentences, but in this case, the material following the semicolon is a fragment. You fix both points simultaneously. I would suggest eliminating the semicolon and trying, "For this step, the bioconcentration factor (BCF) ranges from 10³ – 10⁷, which is several orders of magnitude . . ."

Paragraph 2 does a great job introducing both terminology and the variable effects of DOM in the literature.

Results and Discussion

The PCA results are much more clearly discussed in this revision.

Line 136 - 137. I am not quite following the units of the MeHg binding capacity. You have, "The MeHg binding capacity of the extracted DOM is expressed as the thiol content per carbon of the DOM (RSH/DOC, mol/g; here forward referred to as the DOM binding capacity) . . ." Do you mean that DOM is expressed as the thiol content per carbon MASS of the DOM?

Line 139 and lines 155-157. You have defined both DOM binding capacity and the concentration of the thiols associated with the DOM (DOM-RSH). Can you qualitatively explain the difference in these parameters when you introduce them/why you would want to look at one vs. the other. How is it significant (or is it?) that the DOM binding capacity varies by a factor of 10, but DOM-RSH varies by a factor of 40? Results from both parameters seem to lead to the same conclusion, mainly that total DOC concentration is not a great proxy for MeHg binding.

Lines 140 – 141. Here you have the DOC measured in g/L, but elsewhere in the paper (e.g., Fig. 4), you report DOC in terms of μM .

Line 175- 176. This is an exciting and clearly articulated application of your paper – mainly, that researchers can use your RSH/DOC ratios, calculated for each ecosystem, to guesstimate how much MeHg is bound to thiols at sites where RSH/DOC has not been determined (which, of course,

is most sites).

Line 188. I suggest a paragraph break beginning with, "We determined MeHg-DOM stability constants . . ." One paragraph to describe your hypothesis and one paragraph to describe your results would be easier to read.

Line 266. I wonder to what extent the poor fit of the model in Fig. 4b is due to the large error bars for the River DOM at the low concentration. This is the same data point that was miscalculated in the first version of this manuscript. From the methods, the cellular uptake experiment was run in triplicate. I am curious about what the data points look like – all three spread out? Or one far from the others? A more detailed discussion of the error bars would be appropriate, especially given that other error bars (such as the Marsh DOM) are much smaller. Also, some points do not have error bars at all. This is confusing because the methods say that these experiments were run in triplicate. Please address that here or in the methods.

Line 269. How was the equation shown on the panels in Figure 4 developed? Was the VCF for each of the triplicates averaged so that the equation shows the fit to six data points? Or did each replicate contribute individually (creating 18 data points)? If the latter, then points where the treatments were run in triplicate would have had more weight – that would be problematic.

Paragraph beginning on line 271. This is a nice summary.

Line 349. Nice application of your hypothesis to explain the odd results in the Pickhardt and Fisher study.

Paragraph beginning on line 374. Nice connections with how autochthonous and allochthonous DOM could affect MeHg availability under various global change scenarios.

Methods

Line 591. What is the "C"? Does it mean DOC?

Line 592. Don't start a sentence with a number.

Line 596. Try a new paragraph to describe how the filters were digested and analyzed.

Tables and Figures

Table 1.

You could give the reader just a little more help in the caption by writing out that the DOM-RSH shows the concentration of DOM associated thiols. Also, RSH/DOC shows the DOM binding capacity, measured by LC-MS/MS, and it shows the thiol content per mass of the DOC.

Figure 4.

The previous version of this figure graphed the inverse function of the parameters on the x-axis (e.g., 1/DOM-RSH or 1/DOC). This version is much better.

The title for this figure is a bit long and clunky. I suggest that you end the title for the figure in the middle of line 829, so that the entire title is "Phytoplankton MeHg volume concentration factors (VCF) for a marine diatom, *Thalassiosira pseudonana*, under varied dissolved organic matter concentrations." Then, use the caption to provide more details about the panels, such as "Panel A shows the concentration of the thiols associated with the DOM (DOM-RSH). Panel B shows . . ."

Also state in the caption what the error bars are showing (standard deviation? Standard error?).

Figure 5.

All of these circles will be indistinguishable once printed in black and white. Can you use some other shapes, such as squares and triangles? Or at least some circles with hash marks? Similarly, in panel 2, use hash marks to distinguish the bars so that this figure is legible in black and white.

In the previous figure, you used A, B, and C for the panels, but this figure uses 1, 2, and 3. Be consistent.

Figure 5 Panel 1 looks like the one presented in Figure S5, Case A, but the conclusion from that discussion seemed to be that Case A was not complex enough, and Case B was probably better. Why show Case A in the take-home figure from your paper, if you think Case B is a better representation of what is happening?

Panel 2. The reader has to study the figure carefully and read the caption to see that DOC and DOM-RSH are not decreasing proportionately across the continuum of sites. The caption talks about the RSH/DOC ratio, but it is not graphed. In the response to my previous comment asking to graph the RSH/DOC ratio, the authors noted that adding RSH/DOC to the figure might make it difficult to read. Point noted, but is there another way to highlight that the change in DOC and the change in DOM-RSH are not proportional? Maybe add little pie charts? It would be good to really drive home the point visually that DOC is not a good proxy for DOM-RSH.

Supplemental Info

Table S1 (Supplementary dataset). Thanks for providing all of the data. Don't forget to change salinity to PSU, consistent with your other units in the main body of the manuscript. I would suggest that you also change DOC to of μM , again to be consistent with Figure 4 in the main manuscript.

Reviewer #3 (Remarks to the Author):

This manuscript presents novel data on the importance of DOM associated thiols on methylmercury bioavailability in aquatic ecosystems. The combination of experimental and field measurements provides compelling evidence for a gradient in methylmercury bioavailability along a terrestrial-marine aquatic continuum.

I think the authors have adequately addressed previous reviewer comments for the most part. However, there are a few outstanding points that should be addressed to better present the findings in a broader context of methylmercury cycling.

Main comments:

1) The accumulation of MeHg in phytoplankton is the net effect of various processes and environmental conditions, one of which is MeHg bioavailability. Aqueous MeHg concentration is another key factor and the importance of MeHg production in the aquatic ecosystem needs to be acknowledged (i.e. to remind the reader of other competing drivers). For example, the conceptual model of Figure 5 may give the impression to the reader that the coastal shelf has the highest MeHg concentrations of phytoplankton and marshes the lowest. The figure heading states "A conceptual figure illustrating that the accumulation of MeHg increases in the order: marsh < river... etc". The statement is misleading because the conceptual model illustrates bioavailability, not accumulation. This study did not measure phytoplankton accumulation of MeHg at sites along the terrestrial-marine continuum, where aqueous MeHg concentrations varied among sites (Table S1). It is possible that high aqueous MeHg concentrations in marshes (where mercury methylation rates can be very high) may lead to greater phytoplankton accumulation of MeHg in those systems despite lower bioavailability. In other words, the dampening effect of DOM-thiols on bioavailability may be secondary when the aqueous MeHg concentrations are high. This point should be made in the text (e.g., perhaps in the paragraph of lines 352-356) and the figure heading should be revised.

2) Similarly, the magnitude of the effect of DOM-thiols on MeHg bioavailability is difficult to interpret when phytoplankton MeHg accumulation is presented as volume concentration factors (Figure 4). I don't disagree with this approach – it is a valid method for evaluating MeHg uptake in unicellular organisms. However, additional data in the supplementary information would be helpful. In particular, the reporting of phytoplankton MeHg concentrations (on a mass basis, e.g. ng/g) from the uptake experiment would provide more context for the effect size of DOM-thiols on MeHg

accumulation. Given that the aqueous MeHg concentration in the experiment was the same across treatments, the differences between treatments could be evaluated.

3) In the Implications section, it is unclear what is the basis for the statement that "This effect [on MeHg bioavailability and bioaccumulation] is expected to be large at low-to-moderate nanomolar-level concentrations..." (line 386-387). There may be a greater effect on bioavailability but that does not necessarily translate to a large effect on bioaccumulation because of the potential importance of other environmental factors (e.g., aqueous MeHg concentration). This study presents novel and important research on the mechanisms of DOM effects of MeHg bioavailability; however, caution is warranted regarding over-reaching statements on the implications of the findings for bioaccumulation.

4) Reviewer 2 makes the point that the study focuses on high molecular weight DOM. I feel that the authors did not adequately address this point in their revisions. There has been considerable research over the last decade investigating the effects of low-molecular weight thiols on the uptake of mercury in microbes, including findings of enhanced bioavailability (e.g., <https://doi.org/10.1021/acs.est.5b00676>; <https://doi.org/10.1371/journal.pone.0138333>; <https://doi.org/10.1038/ngeo412>; <https://pubs.acs.org/doi/10.1021/acs.est.8b02709>; <https://doi.org/10.1073/pnas.1105781108>). While much of this research has focused on uptake of inorganic Hg in bacteria and arguably may not directly relevant to MeHg uptake by phytoplankton, a brief statement should be added to acknowledge this work and to present the findings from the present study in the broader context of research about thiol effects on mercury bioavailability (e.g. DOM molecular weight). Recent findings from Li et al. 2022 (Environ Pollution, <https://doi.org/10.1016/j.envpol.2022.120111>) showed the molecular size of thiols was a critical factor for effects on MeHg uptake by phytoplankton.

Minor comments:

Line 346-351. The discussion on an anomalous site in a different study by Pickhardt and Fisher and the possibility that the explanation is due to elevated DOM-RSH is highly speculative. It does not lend support to the importance of DOM-RSH on MeHg bioavailability and should be removed.

Line 366-368. This statement is a bit misleading because it seems to suggest that a number of field studies have found higher seston MeHg concentrations in marine environments along a terrestrial-marine aquatic continuum. The terrestrial study for MeHg in stream seston (reference 19) did not look at marine environments and therefore is not relevant for a statement about effects of terrestrial vs marine DOM. Similarly, the references 52 and 53 did not sample sites along a gradient of terrestrial DOM influences in the marine environment. A revision is needed here.

Line 585. More information is needed in the supplemental information to describe the methods used for measurement of MeHg cellular uptake. Measurement of MeHg in seston or phytoplankton can be challenging due to low cellular MeHg concentrations and the low amount of sample mass. Confirmation is requested that MeHg concentrations of phytoplankton samples were well above analytical detection in all samples. Details on how cell counts and cell volumes were determined should also be added.

Response to reviewers

We are grateful for the overall positive feedback and the specific remarks from the reviewers. We have carefully revised the manuscript following the requests and recommendations from the reviewers. In our below responses (in blue font), the line numbers refer to the revised manuscript file with track changes indicated.

Reviewer #1 (Remarks to the Author):

Overall

This paper has undergone substantial revisions since the last version. The authors have made every effort to address the reviewers' comments and have explained their changes in the author response. Great job with such a thorough and thoughtful response. The revisions have resulted in a manuscript that is much easier to read and has addressed some problems from the last version, such as an incorrectly calculated VCF value. More minor problems, such as missing units, or too many acronyms, have also been addressed. The figures have been improved. For example, the x-axis on Fig. 4 is much more easily interpretable.

The paper has clearly articulated its key finding, mainly that DOC is not a great proxy for the thiol concentration in the DOM, and the latter (DOM-RSH) is key to predicting MeHg bioavailability. This finding helps resolve a long-standing conundrum about why DOM has been found to have variable effects on phytoplankton MeHg accumulation. The limitation of this study is that Fig. 4 uses only 6 data points to show that DOM-RSH is a better predictor for VCF than DOC. But I suppose that future studies will fill in with more data, and this paper will get the ball rolling! Below, I have some comments on specific points in the manuscript. At this point, my comments are mostly minor clarifications, not substantial changes. Overall, nice work!

We appreciate the positive reviewer comments, and are grateful for the improvements of the manuscript that were made possible based on the first round of review.

Abstract

The abstract now does a better job of highlighting key findings and specific magnitudes, such as in the sentence, "Across 20 study sites, DOM-RSH concentrations decreased 40-fold from terrestrial to marine environments whereas dissolved organic carbon (DOC), the typical proxy for MeHg binding sites in DOM, only had a 5-fold decrease." The direction of the effect in relationship to the terrestrial-marine continuum is also clear from the abstract.

Thank you.

Introduction

Line 35. The use of the reference number in superscript right next to 10⁷ is potentially confusing. In addition, the use of the semicolon is not correct. A semicolon is used to connect two complete sentences, but in this case, the material following the semicolon is a fragment. You fix both points simultaneously. I would suggest eliminating the semicolon and trying, "For this step, the bioconcentration factor (BCF) ranges from 10³ – 10⁷, which is several orders of magnitude . . ."

We agree and have revised the text to:

Line 35:

"For this step, bioconcentration factors range from 3 to 7 log units⁶..."

Paragraph 2 does a great job introducing both terminology and the variable effects of DOM in the literature.

Thank you.

Results and Discussion

The PCA results are much more clearly discussed in this revision.

Thank you.

Line 136 - 137. I am not quite following the units of the MeHg binding capacity. You have, "The MeHg binding capacity of the extracted DOM is expressed as the thiol content per carbon of the DOM (RSH/DOC, mol/g; here forward referred to as the DOM binding capacity) . . ." Do you mean that DOM is expressed as the thiol content per carbon MASS of the DOM?

Yes, it was initially expressed as thiol content per gram carbon of the DOM. However, based on a following comment, we adjusted all units to be mol based. Therefore, the sentence was modified to:

Line 136-137:

"The MeHg binding capacity of the extracted DOM is expressed as the thiol content per mol carbon of the DOM (RSH/DOC, $\mu\text{mol}/\text{mol}$; here forward referred to as the DOM binding capacity) . . ."

Line 139 and lines 155-157. You have defined both DOM binding capacity and the concentration of the thiols associated with the DOM (DOM-RSH). Can you qualitatively explain the difference in these parameters when you introduce them/why you would want to look at one vs. the other. How is it significant (or is it?) that the DOM binding capacity varies by a factor of 10, but DOM-RSH varies by a factor of 40? Results from both parameters seem to lead to the same conclusion, mainly that total DOC concentration is not a great proxy for MeHg binding.

The DOM binding capacity, as we define it, describes an inherent property of the DOM, whereas the *in situ* binding capacity is a characteristic of the site. In the environment, MeHg bioavailability is determined by the *in situ* binding capacity, which is determined by both the site specific DOM binding capacity and the DOC concentration. It is important to specifically identify the variability in DOM binding capacities at different sites as it explains why DOC alone is a poor proxy for *in situ* MeHg binding capacities. We described the qualitative differences between the two parameters in line 140-141, and have now further clarified this statement by replacing DOM-RSH concentration and RSH/DOC ratio with the respective binding capacity definitions

Line 140-141: *"The *in situ* binding capacity was determined by multiplying the DOM binding capacity by the local DOC concentration (mol/L)."*

We hope that the distinction between, and usefulness of, the two binding capacity terms thereby are clear.

Lines 140 – 141. Here you have the DOC measured in g/L, but elsewhere in the paper (e.g., Fig. 4), you report DOC in terms of μM .

We agree that this is inconsistent and we have revised all DOC and DOM-RSH units to molar units throughout the manuscript.

Line 175- 176. This is an exciting and clearly articulated application of your paper – mainly, that researchers can use your RSH/DOC ratios, calculated for each ecosystem, to guesstimate how much MeHg is bound to thiols at sites where RSH/DOC has not been determined (which, of course, is most sites).

Thank you.

Line 188. I suggest a paragraph break beginning with, “We determined MeHg-DOM stability constants . . .” One paragraph to describe your hypothesis and one paragraph to describe your results would be easier to read.

Thank you, we agree and have made the change as suggested.

Line 266. I wonder to what extent the poor fit of the model in Fig. 4b is due to the large error bars for the River DOM at the low concentration. This is the same data point that was miscalculated in the first version of this manuscript. From the methods, the cellular uptake experiment was run in triplicate. I am curious about what the data points look like – all three spread out? Or one far from the others? A more detailed discussion of the error bars would be appropriate, especially given that other error bars (such as the Marsh DOM) are much smaller. Also, some points do not have error bars at all. This is confusing because the methods say that these experiments were run in triplicate. Please address that here or in the methods.

We agree with the reviewer that these aspects need to be clearer. The poor fit in Fig. 4b is not caused by the large error bar for the river sample with lowest DOM content, in fact the data points for all replicates are fairly evenly distributed, see below figures where all replicates are plotted.

We have now added both the VCF and MeHg concentration data for each replicate to Supplementary Table S1b so that all data is available for the reader.

Further, all data points have error bars in Fig. 4 but in some cases the error bars are too small to be visible. It is expected that the variability will increase at very low concentrations of DOM-RSH and DOC. We have added the following text to the Fig. 4 legend:

Line 820-822:

“Error bars represent propagated uncertainty of VCFs based on uncertainties in measured MeHg concentrations in plankton and water (n = 3, except for site SI with lowest DOM-RSH and DOC concentration where n=2) and are smaller than visible in some cases. Replicate VCF data are reported in Table S1.”

Line 269. How was the equation shown on the panels in Figure 4 developed? Was the VCF for each of the triplicates averaged so that the equation shows the fit to six data points? Or did each replicate contribute individually (creating 18 data points)? If the latter, then points where the treatments were run in triplicate would have had more weight – that would be problematic.

All VCF conditions were run in triplicates (one data point missing for site SI with lowest DOM-RSH and DOC concentration) and the equation was fitted to the average data points for each of the six conditions. All data points thus have the same weight.

Paragraph beginning on line 271. This is a nice summary.

Thank you.

Line 349. Nice application of your hypothesis to explain the odd results in the Pickhardt and Fisher study.

Thank you. As noted by reviewer #3, our interpretation of the Pickhardt and Fisher study is partly speculative (as we note in the first sentence of that paragraph, line 339) which needs to be clearly acknowledged in the discussion. We do, however, think that it is interesting to highlight this study as the results are anomalous and our findings offer a potential explanation. We have revised the text to better acknowledge the uncertainty and avoid placing too much weight on our proposed explanation.

Line 352-354:

“Based on our results, it can be hypothesized that the marsh site had elevated DOM-RSH concentration and therefore suppress uptake to a greater extent than DOC concentrations alone would predict. This effect, however, remains to be tested for the specific sites.”

Paragraph beginning on line 374. Nice connections with how autochthonous and allochthonous DOM could affect MeHg availability under various global change scenarios.

Thank you.

Methods

Line 591. What is the “C”? Does it mean DOC?

Yes, and it was changed to DOC for clarity. (Line 593)

Line 592. Don’t start a sentence with a number.

We agree and have rearranged the sentence, line 606-607: “After the equilibration period, 2.4×10^7 cells of *Thalassiosira pseudonana* were added to each 200 mL flask (1.2×10^5 cells/mL)...”

Line 596. Try a new paragraph to describe how the filters were digested and analyzed.

We believe it is better to keep this text as one paragraph not to break the text flow in a way which may appear confusing for the reader.

Tables and Figures

Table 1.

You could give the reader just a little more help in the caption by writing out that the DOM-RSH shows the concentration of DOM associated thiols. Also, RSH/DOC shows the DOM binding capacity, measured by LC-MS/MS, and it shows the thiol content per mass of the DOC.

We agree that more information in the Table 1 caption was beneficial. It has been revised.

Lines 781-786:

“Classification and location of the study sites and selected key parameters of the sites and extracted dissolved organic matter (DOM) samples. The site locations are within the United States and shown in Fig. 1. System types were determined based on individual site features and local salinity, as defined in the text. The carbon (C), nitrogen (N), and sulfur (S) percentages were calculated by weight. Site specific thiol concentrations (DOM-RSH in water) were calculated by multiplying the thiol (RSH) to dissolved organic carbon (DOC) ratio ($\mu\text{mol/mol}$) by the in situ DOC concentration.”

Figure 4.

The previous version of this figure graphed the inverse function of the parameters on the x-axis (e.g., $1/\text{DOM-RSH}$ or $1/\text{DOC}$). This version is much better.

Thank you, we agree and believe that the new version will be easier to interpret for the reader.

The title for this figure is a bit long and clunky. I suggest that you end the title for the figure in the middle of line 829, so that the entire title is “Phytoplankton MeHg volume concentration factors (VCF) for a marine diatom, *Thalassiosira pseudonana*, under varied dissolved organic matter concentrations.” Then, use the caption to provide more details about the panels, such as “Panel A shows the concentration of the thiols associated with the DOM (DOM-RSH). Panel B shows . . .”

We agree with the remark and have revised the figure legend text accordingly.

Line 816-822:

*“Phytoplankton MeHg volume concentration factors (VCF) for a marine diatom, *Thalassiosira pseudonana*, for two contrasting DOM types, one marsh site (SI) and one river site (OR). Panel a) depicts average VCFs versus the concentration of thiol ligands in the experimental matrix. Panel b) shows the same VCF information but plotted against the concentration of dissolved organic carbon (DOC) used in*

the experiments. Error bars represent propagated uncertainty of VCFs based on uncertainties in measured MeHg concentrations in plankton and water ($n = 3$, except for site SI with lowest DOM-RSH and DOC concentration where $n=2$) and are smaller than visible in some cases. Replicate VCF data are reported in Table S1.”

Also state in the caption what the error bars are showing (standard deviation? Standard error?).

The error bars represent the propagated uncertainty of measured MeHg concentrations in plankton cells and the assay buffer solution. This information is now included in the figure caption as detailed in a comment above.

Figure 5.

All of these circles will be indistinguishable once printed in black and white. Can you use some other shapes, such as squares and triangles? Or at least some circles with hash marks? Similarly, in panel 2, use hash marks to distinguish the bars so that this figure is legible in black and white.

To improve the legibility of this figure in color or black and white, we have increased the size differences between the circles (The MeHg circles are now larger) and included a picture legend. We believe there is enough contrast between the colored bars in panel b to be distinguishable in black and white without hash marks. We felt as though a patterned fill was too busy.

In the previous figure, you used A, B, and C for the panels, but this figure uses 1, 2, and 3. Be consistent. **We have revised to a), b), and c) for the three panels to be consistent for all figures in the manuscript.**

Figure 5 Panel 1 looks like the one presented in Figure S5, Case A, but the conclusion from that discussion seemed to be that Case A was not complex enough, and Case B was probably better. Why show Case A in the take-home figure from your paper, if you think Case B is a better representation of what is happening?

We agree with the reviewer's remark. We also want to keep the figure general and be open for both cases A and B. We have therefore revised the figure to include both cases and added the following text in the figure legend to panel a):

Line 831-850:

"Figure 5. Conceptual diagram depicting that dissolved organic matter's thiol ligand content (DOM-RSH), not carbon content (DOC), dictates MeHg bioavailability to plankton cells across the terrestrial-marine aquatic continuum. Panel a shows how the DOM-RSH concentration may control uptake by a direct equilibrium partitioning of MeHg between DOM-RSH and plankton cells (case A) and/or by an equilibrium between MeHg(DOM-RS) and MeHgCl complexes internalized by cells with species-specific rates (case B)....."

Panel 2. The reader has to study the figure carefully and read the caption to see that DOC and DOMRSH are not decreasing proportionately across the continuum of sites. The caption talks about the RSH/DOC ratio, but it is not graphed. In the response to my previous comment asking to graph the RSH/DOC ratio, the authors noted that adding RSH/DOC to the figure might make it difficult to read. Point noted, but is there another way to highlight that the change in DOC and the change in DOM-RSH are not proportional? Maybe add little pie charts? It would be good to really drive home the point visually that DOC is not a good proxy for DOM-RSH.

We agree with the reviewer's remark and to improve clarity we have changed to logarithmic scale for the DOC and DOM-RSH concentrations and we have added the numerical values of the RSH/DOC ratios.

Figure 5. a) Conceptual diagram depicting that dissolved organic matter's thiol ligand content (DOM-RSH), not carbon content (DOC), dictates MeHg bioavailability to plankton cells across the terrestrial-marine aquatic continuum. Panel a shows how the DOM-RSH concentration may control uptake by a direct equilibrium partitioning of MeHg between DOM-RSH and plankton cells (case A) and/or by an equilibrium between MeHg(DOM-RS) and MeHgCl complexes internalized by cells with species-specific rates (case B). In panel b, DOM-RSH (nM) and DOC (μM) concentrations across the terrestrial-marine aquatic continuum are shown using system-averaged bar plots (full dataset presented in Table 1). The numerical values above the bars are the system specific RSH/DOC (nM/ μM) ratios. The substantial and systematic change in DOM-RSH concentrations across the continuum are driven by a decrease in both RSH/DOC ratios and DOC concentrations, resulting in a larger range of observed DOM-RSH compared to DOC. The implications of panel a and b are conceptualized in panel c. Namely, that the bioavailability of MeHg (depicted as volume concentration factors (VCF) in plankton) increases in the order: marsh < river \approx estuary < shelf because of the decreasing DOM-RSH concentration in water."

Supplemental Info

Table S1 (Supplementary dataset). Thanks for providing all of the data. Don't forget to change salinity to PSU, consistent with your other units in the main body of the manuscript. I would suggest that you also change DOC to of μM , again to be consistent with Figure 4 in the main manuscript.

Thank you. We have changed the salinity and DOC as suggested. Further, the RSH/DOC ratios are presented as a mol ratio to match the text.

Reviewer #3 (Remarks to the Author):

This manuscript presents novel data on the importance of DOM associated thiols on methylmercury bioavailability in aquatic ecosystems. The combination of experimental and field measurements provides compelling evidence for a gradient in methylmercury bioavailability along a terrestrial-marine aquatic continuum.

I think the authors have adequately addressed previous reviewer comments for the most part. However, there are a few outstanding points that should be addressed to better present the findings in a broader context of methylmercury cycling.

We appreciate the overall positive comments and are grateful for the specific remarks. We have revised the manuscript accordingly.

Main comments:

1) The accumulation of MeHg in phytoplankton is the net effect of various processes and environmental conditions, one of which is MeHg bioavailability. Aqueous MeHg concentration is another key factor and the importance of MeHg production in the aquatic ecosystem needs to be acknowledged (i.e. to remind the reader of other competing drivers). For example, the conceptual model of Figure 5 may give the impression to the reader that the coastal shelf has the highest MeHg concentrations of phytoplankton and marshes the lowest. The figure heading states “A conceptual figure illustrating that the accumulation of MeHg increases in the order: marsh < river... etc”. The statement is misleading because the conceptual model illustrates bioavailability, not accumulation. This study did not measure phytoplankton accumulation of MeHg at sites along the terrestrial-marine continuum, where aqueous MeHg concentrations varied among sites (Table S1). It is possible that high aqueous MeHg concentrations in marshes (where mercury methylation rates can be very high) may lead to greater phytoplankton accumulation of MeHg in those systems despite lower bioavailability. In other words, the dampening effect of DOM-thiols on bioavailability may be secondary when the aqueous MeHg concentrations are high. This point should be made in the text (e.g., perhaps in the paragraph of lines 352-356) and the figure heading should be revised.

We agree with the reviewer that the term “bioavailability”, and not “accumulation”, should be used in this context and we have made this correction in the figure legend. In response to this remark and to remark 4) below, we have further revised the following text to point out the importance of the total MeHg concentration in water and the potential importance of molecular structure of thiol compounds:

Line 355-365:

“There are additional factors that contribute to differences in MeHg BCF across systems such as phytoplankton abundance, community composition, cellular size distribution, and dissolved MeHg concentrations^{12,25}. Further, laboratory studies have shown that the cellular uptake rate of MeHg-thiol complexes by phytoplankton^{12,23}. (and similarly the uptake of inorganic Hg-thiol complexes by bacteria^{54,55}) can differ depending on the chemical structure of the DOM-RSH compound (e.g. molecular weight and potential for chelation effects). More work is needed to clarify to what extent the molecular composition of thiols varies in natural DOM, and if this is a contributing controlling factor for MeHg bioavailability in the environment⁵⁶. Our results highlight the significant improvement that quantifying DOM-RSH can

have when parameterizing MeHg BCF results from natural systems instead of relying on DOC as the binding ligand proxy.”

2) Similarly, the magnitude of the effect of DOM-thiols on MeHg bioavailability is difficult to interpret when phytoplankton MeHg accumulation is presented as volume concentration factors (Figure 4). I don't disagree with this approach – it is a valid method for evaluating MeHg uptake in unicellular organisms. However, additional data in the supplementary information would be helpful. In particular, the reporting of phytoplankton MeHg concentrations (on a mass basis, e.g. ng/g) from the uptake experiment would provide more context for the effect size of DOM-thiols on MeHg accumulation. Given that the aqueous MeHg concentration in the experiment was the same across treatments, the differences between treatments could be evaluated.

We agree that phytoplankton MeHg concentrations data is useful to include and we have added this data to the Supporting information, Table S1b. As expected for this uptake experiment, the data for MeHg concentration in plankton shows the same pattern as the VCF data but is slightly more scattered as (the slight) variations in dissolved MeHg concentration is not taken into account.

3) In the Implications section, it is unclear what is the basis for the statement that “This effect [on MeHg bioavailability and bioaccumulation] is expected to be large at low-to-moderate nanomolar-level concentrations...” (line 386-387). There may be a greater effect on bioavailability but that does not necessarily translate to a large effect on bioaccumulation because of the potential importance of other environmental factors (e.g., aqueous MeHg concentration). This study presents novel and important research on the mechanisms of DOM effects of MeHg bioavailability; however, caution is warranted regarding over-reaching statements on the implications of the findings for bioaccumulation.

We agree with the reviewer that also in this context it is important to be clear about the fact that our discussion concerns bioavailability and not bioaccumulation. We have revised the text accordingly.

Line 400:

“Our study highlights that any environmental process causing an increase in the molar concentration of DOM-RSH will decrease MeHg bioavailability.”

4) Reviewer 2 makes the point that the study focuses on high molecular weight DOM. I feel that the authors did not adequately address this point in their revisions. There has been considerable research over the last decade investigating the effects of low-molecular weight thiols on the uptake of mercury in microbes, including findings of enhanced bioavailability (e.g., <https://doi.org/10.1021/acs.est.5b00676>; <https://doi.org/10.1371/journal.pone.0138333>; <https://doi.org/10.1038/ngeo412>; <https://pubs.acs.org/doi/10.1021/acs.est.8b02709>; <https://doi.org/10.1073/pnas.1105781108>). While much of this research has focused on uptake of inorganic Hg in bacteria and arguably may not directly relevant to MeHg uptake by phytoplankton, a brief statement should be added to acknowledge this work and to present the findings from the present study in the broader context of research about thiol effects on mercury bioavailability (e.g. DOM molecular weight). Recent findings from Li et al. 2022 (Environ Pollution, <https://doi.org/10.1016/j.envpol.2022.120111>) showed the molecular size of thiols was a critical factor for effects on MeHg uptake by phytoplankton.

We acknowledge the remark and realize that our previous revisions to the manuscript were not fully clear regarding this point and we have now further clarified two, partly different, aspects.

First, the main point in our response to reviewer's #2 remark was that our study does NOT focus on high molecular weight DOM. The used PPL-SPE approach efficiently extracts a broad range of compounds and do not discriminate based on molecular size per se. However, while also highly hydrophilic compounds (which often also have comparably small weight) are extracted, the recovery of such compounds can be lower than average. This needs to be considered but there is no reason to believe that the SPE extraction impacts the findings or conclusions in our study. We have clarified this aspect in the revised manuscript text as follows:

Line 237-249:

"Hypothetically, the results of the kinetic exchange tests could be influenced by the use of extracted DOM if the molecular composition of a water sample impacts the extraction yields of specific compounds. While the PPL-SPE used in this study captures a broad range of compounds and retains overall differences in the molecular composition of the DOM, recent studies suggest that small hydrophilic molecules can be partly lost from the sorbent during the extraction process³¹. Such compounds form the most bioavailable and easily exchanged MeHg-thiol complexes^{12,23}. In theory, if the recovery of such compounds were incomplete in the DOM extraction, the in situ turnover for MeHg ligand exchange might be even faster than the <1-3 min we calculated. Therefore, our conclusion that MeHg ligand exchange kinetics among DOM-RSH compounds is fast and not a major control of MeHg bioavailability would still be valid, or even amplified, if there were partial losses of small hydrophilic thiols in the SPE isolation."

Second, we agree with the reviewer that previous studies on the effects of low-molecular weight thiols on the uptake of Hg in microbes should be properly acknowledged. As noted by the reviewer, laboratory studies have shown that the cellular uptake rate of Hg-thiol complexes can differ depending on the chemical structure of the thiol compound. This has been demonstrated both for the uptake of MeHg-thiol complexes by phytoplankton and of inorganic Hg by bacteria. We note that these studies suggest that the effect of thiols is partly related to size/weight of the thiol compounds but also that the effects are more intricate and that thiols forming chelation-type bonds with Hg suppress uptake compared to thiols which do not form such structures. Indeed, for the uptake of inorganic Hg, largely different uptake rates have been observed for Hg-thiol complexes with very similar molecular weight/size. The differences in uptake among various Hg-thiol complexes further seem to be smaller for uptake of MeHg by phytoplankton compared to uptake of inorganic Hg by bacteria (compare Lee and Fischer 2017 and Skrobonja et al. 2019 for MeHg versus Schaefer et al. 2011 and Adediran et al. 2019 for inorganic Hg). Regarding the study by Li et al. 2022, they reported lowest uptake for the DOM with the smallest thiol compounds, which is apparently contradictory to previous studies and to general theory on metal uptake by microbes. It is, however, not clear from the study what causal mechanism underlies this observation since the DOM with the smallest thiol compounds also had the largest stability constant for MeHg binding (although all reported stability constants were very low compared to what is expected for MeHg-thiol complexes). As noted in previous studies concerning uptake of MeHg (Skrobonja et al. 2019) and of inorganic Hg (Adediran et al. 2019), it remains to characterize the molecular composition of thiol compounds in natural DOM and to clarify if composition variations are of importance for Hg bioavailability in the environment.

As suggested by the reviewer, we have added brief statements to acknowledge these studies and provide further context regarding research about thiol effects on mercury bioavailability.

Line 355-365:

“There are additional factors that contribute to differences in MeHg BCF across systems such as phytoplankton abundance, community composition, cellular size distribution, and dissolved MeHg concentrations^{12,25}. Further, laboratory studies have shown that the cellular uptake rate of MeHg-thiol complexes by phytoplankton^{12,23}. (and similarly the uptake of inorganic Hg-thiol complexes by bacteria^{54, 55}) can differ depending on the chemical structure of the DOM-RSH compound (e.g. molecular weight and potential for chelation effects). More work is needed to clarify to what extent the molecular composition of thiols varies in natural DOM, and if this is a contributing controlling factor for MeHg bioavailability in the environment⁵⁶. Our results highlight the significant improvement that quantifying DOM-RSH can have when parameterizing MeHg BCF results from natural systems instead of relying on DOC as the binding ligand proxy.”

Minor comments:

Line 346-351. The discussion on an anomalous site in a different study by Pickhardt and Fisher and the possibility that the explanation is due to elevated DOM-RSH is highly speculative. It does not lend support to the importance of DOM-RSH on MeHg bioavailability and should be removed.

We agree that our interpretation of the Pickhardt and Fisher study is partly speculative (as we note in the first sentence of that paragraph, line 339) which needs to be clearly acknowledged in the discussion. We do, however, also think that it is interesting to highlight this study as the results are anomalous and our findings offer a potential explanation. We have revised the text to better acknowledge the uncertainty and avoid placing too much weight on our proposed explanation.

Line 352-354:

“Based on our results, it can be hypothesized that the marsh site had elevated DOM-RSH concentration and therefore suppress uptake to a greater extent than DOC concentrations alone would predict. This effect, however, remains to be tested for the specific sites.”

Line 366-368. This statement is a bit misleading because it seems to suggest that a number of field studies have found higher seston MeHg concentrations in marine environments along a terrestrial-marine aquatic continuum. The terrestrial study for MeHg in stream seston (reference 19) did not look at marine environments and therefore is not relevant for a statement about effects of terrestrial vs marine DOM. Similarly, the references 52 and 53 did not sample sites along a gradient of terrestrial DOM influences in the marine environment. A revision is needed here.

We agree that this statement could be misinterpreted in the way the reviewer describes. We have revised the text to clarify that the described pattern (higher seston MeHg concentrations in systems with higher levels of marine DOM over terrestrial DOM) emerges by comparing results from several studies each focusing on isolated parts of the continuum.

Line 378-382:

“A comparison of the results from field observations in terrestrial¹⁹, estuarine, and/or marine aquatic systems^{17,52,53} show higher MeHg concentrations in seston from systems with higher levels of marine DOM over terrestrial DOM. Such trends have also been observed in laboratory uptake experiments¹⁸.”

Line 585. More information is needed in the supplemental information to describe the methods used for measurement of MeHg cellular uptake. Measurement of MeHg in seston or phytoplankton can be challenging due to low cellular MeHg concentrations and the low amount of sample mass. Confirmation is requested that MeHg concentrations of phytoplankton samples were well above analytical detection in all samples. Details on how cell counts and cell volumes were determined should also be added.

More information has been provided in the “MeHg Cellular Uptake Experiment” portion of the methods section to clarify the uptake results.

We now confirm that the MeHg concentration in the plankton digests were above the analytical detection limit:

Line 618-619: *“The MeHg concentration was above the analytical detection limit of 0.004 ng/L (range = 0.018-0.532 ng/L), and analytical duplicates had RSDs of $15 \pm 6.0\%$ ($n = 4$) for all the particulate digests.”*

We also included more information regarding the dissolved MeHg concentrations:

Line 621-623: *“All samples were above the analytical detection limit (0.35-1.05 ng/L) but were not run in duplicate. Treatment replicates, however, had an average RSD of $11 \pm 5.3\%$ and spike recoveries of $97 \pm 24\%$.”*

It is now clarified that cell counts and size were determined by a coulter counter:

Line 606-608: *“After the equilibration period, 2.4×10^7 cells of *Thalassiosira pseudonana* were added to each 200 mL flask (1.2×10^5 cells/mL), determined by counting the cell density of the stock culture with a Coulter Multisizer IIe,…”*

Line 612: *“The final cell count of the incubated culture was determined using a Coulter Multisizer IIe.”*

Finally, we clarify the cell size and other methods used for the VCF calculation:

Line 623-626: *“The suspended particulate MeHg concentrations were converted to nmol MeHg/cell volume for the VCF calculation. To do so, a cell diameter of $8 \mu\text{m}$ was used ($113.1 \mu\text{m}^3$) based on the average cell size from the Coulter counter, and the initial MeHg cellular concentrations were subtracted from the final cellular concentrations.”*